# A conserved PI(4,5)P2–binding domain is critical for immune regulatory function of DOCK8

Tetsuya Sakurai[1,*], Mutsuko Kukimoto-Niino[2,*] , Kazufumi Kunimura[1] , Nana Yamane[1], Daiji Sakata[1], Ryosuke Aihara[1], Tomoharu Yasuda[3], Shigeyuki Yokoyama[4] , Mikako Shirouzu[2], Yoshinori Fukui[1] , Takehito Uruno[1]

DOCK8 is a Cdc42-specific guanine-nucleotide exchange factor that is essential for development and functions of various subsets of leukocytes in innate and acquired immune responses. Although DOCK8 plays a critical role in spatial control of Cdc42 activity during interstitial leukocyte migration, the mechanism remains unclear. We show that the DOCK homology region (DHR)-1 domain of DOCK8 binds specifically to phosphatidylinositol 4,5-bisphosphate (PI(4,5)P2) and is required for its recruitment to the plasma membrane. Structural and biochemical analyses reveal that DOCK8 DHR-1 domain consists of a C2 domain-like core with loops creating the upper surface pocket, where three basic residues are located for stereospecific recognition of phosphoinositides. Substitution of the two basic residues, K576 and R581, with alanine abolished PI(4,5)P2 binding in vitro, ablated the ability of DOCK8 to activate Cdc42 and support leukocyte migration in three-dimensional collagen gels. Dendritic cells carrying the mutation exhibited defective interstitial migration in vivo. Thus, our study uncovers a critical role of DOCK8 in coupling PI(4,5)P2 signaling with Cdc42 activation for immune regulation.

## Introduction

To elicit innate and adaptive immune responses, inflammatory cells must efficiently migrate through the complex, physiological environments in the body. Rho family GTPases such as Rho, Rac, and Cdc42 play a central role in cell migration by controlling directionality, protrusive force generation, adhesion to ECM and actomyosin contraction through the regulation of the membrane-cytoskeletal organization (Hall, 1998; Etienne-Manneville & Hall, 2002; Ridley, 2015). Accordingly, Rho GTPases and their regulators play crucial roles in leukocyte development, activation, differentiation, and migration (Heasman & Ridley, 2008; Tybulewicz & Henderson, 2009). Rho GTPases function as a molecular switch cycling between GDP-bound inactive, and GTP-bound active states, conversions of which are catalyzed by two mechanistically distinct classes of regulators: guanine nucleotide exchange factors (GEFs) for activation, and GTPase activating proteins for inactivation (Hall, 1998; Etienne-Manneville & Hall, 2002; Jaffe & Hall, 2005). Therefore, specific localization patterns of GEFs and GTPase-activating proteins, and a molecular network of their regulation underlie the spatiotemporal control of Rho GTPases activities in the cells. However, the precise mechanism is poorly understood.

DOCK8 (dedicator of cytokinesis 8) is a Cdc42-specific GEF predominantly expressed in hematopoietic lineage cells (Ruusala & Aspenström, 2004; Kunimura et al, 2019a). In humans, loss of function mutations of *DOCK8* cause a combined immunodeficiency characterized by recurrent viral infections, severe allergies, autoimmunity, and early-onset malignancy (Engelhardt et al, 2009; Zhang et al, 2009; Su et al, 2019). Studies indicate that DOCK8 is required for normal development and functions of a wide variety of leukocyte subsets in both innate and adaptive immune systems (Biggs et al, 2017; Kearney et al, 2017; Su et al, 2019; Kunimura et al, 2019b). We and others have shown that DOCK8 is essential for interstitial migration of mature DCs (Harada et al, 2012; Krishnaswamy et al, 2015). When leukocytes migrate through a confined, 3D environment of the tissues such as the interstitium of the skin, they exhibit a mode of amoeboid migration following the guidance of chemoattractants, directed by membrane-cytoskeletal protrusions driven by actin polymerization, but independent of strong adhesive interactions with tissues as well as ECM degradation (Friedl & Weigelin, 2008; Lämmermann & Sixt, 2009). This mode of amoeboid migration is adhesion-free, and particularly suited for scanning cellular networks and tissues. Cdc42 plays an essential role in coordination of leading-edge protrusions during interstitial DC migration (Lämmerman et al, 2009). Like Cdc42-deficient DCs, DOCK8-deficient DCs are defective in migration in 3D collagen

[1]Division of Immunogenetics, Department of Immunobiology and Neuroscience, Medical Institute of Bioregulation, Kyushu University, Fukuoka, Japan    [2]Laboratory for Protein Functional and Structural Biology, RIKEN Center for Biosystems Dynamics Research, Yokohama, Kanagawa, Japan    [3]Division of Immunology and Genome Biology, Department of Molecular and Structural Biology, Medical Institute of Bioregulation, Kyushu University, Fukuoka, Japan    [4]RIKEN Cluster for Science, Technology and Innovation Hub, Yokohama, Japan

Correspondence: fukui@bioreg.kyushu-u.ac.jp; uruno@bioreg.kyushu-u.ac.jp
Tomoharu Yasuda's present address is Department of Immunology, Graduate School of Biomedical and Health Sciences, Hiroshima University, Hiroshima, Japan
*Tetsuya Sakurai and Mutsuko Kukimoto-Niino contributed equally to this work

gels (Harada et al, 2012). Whereas DOCK8 deficiency does not affect global Cdc42 activity, Cdc42 activation at the leading-edge membrane is specifically impaired. Similar defect leads to impaired shape integrity of DOCK8 deficient CD8⁺T and NK cells during their migration in the skin (Zhang et al, 2014). We have further shown that DOCK8 links Cdc42 activation to actomyosin dynamics through the association with an adaptor protein LRAP35a during migration of macrophages and DCs (Shiraishi et al, 2017). Still, it remains unclear how DOCK8 controls Cdc42 activity spatially.

The DOCK family of GEFs comprise 11 members, which are classified into four subfamilies: DOCK-A (DOCK1/2/5), -B (DOCK3/4), -C (DOCK6/7/8), and -D (DOCK9/10/11) based on their sequence and functional similarity (Gadea & Blangy, 2014; Laurine & Côté, 2014; Kunimura et al, 2019a). The DOCK family proteins share two evolutionarily conserved domains: DOCK homology region (DHR)-1 and DHR-2. The DHR-2 domain functions as a Rho GEF catalytic domain with specificity of DOCK-A/B for Rac and DOCK-C/D for Cdc42 (Yang et al, 2009; Kulkarni et al, 2011; Hanawa-Suetsugu et al, 2012; Harada et al, 2012). In addition, DOCK6, DOCK7, and DOCK10 are implicated in activation of both Rac and Cdc42, and a recent study has provided structural evidence for the dual specificity of the DOCK7 DHR-2 domain (Kukimoto-Niino et al, 2019). On the other hand, the DHR-1 domain is considered as a phosphoinositide-binding domain because the DHR-1 domain of the DOCK-A subfamily, DOCK1 and DOCK2, binds to phosphatidylinositol-3,4,5-triphosphate (PI(3,4,5)P3) and plays a crucial role in their recruitment to the leading edge of migrating cells in response to extracellular stimuli (Côté et al, 2005; Kunisaki et al, 2006). So far, a single crystal structure of the DOCK1 DHR-1 domain has provided enough detail about its PI(3,4,5)P3 binding mode (Premkumar et al, 2010). However, there is significant diversity in the amino acid sequence within the phosphoinositide-binding pocket among the DOCK family, which precludes the assumption that all the DHR-1 domains bind PI(3,4,5)P3 equally. Recently, a mass spectrometry analysis using immobilized PIP beads has shown that DOCK8 binds to phosphatidylinositol 4,5-bisphosphate (PI(4,5)P2) (Jungmichel et al, 2014); yet, without further biochemical and structural studies, the phosphoinositide-binding specificity of the DOCK8 DHR-1 domain, and its functional significance remain elusive.

In this study, we use biochemical and structural analyses to show that the DOCK8 DHR-1 domain binds specifically to PI(4,5)P2 and identify three critical basic residues within the upper surface pocket. Our docking simulation reveals the mechanism for stereospecific recognition of PI(4,5)P2 by the DOCK8 DHR-1 domain. Through mutational analyses, we show that disruption of the interaction between the DOCK8 DHR-1 domain and PI(4,5)P2 impairs DOCK8 function. Our data, thus, reveal a critical role of the DOCK8 DHR-1 domain in coupling phosphoinositide signaling with the DHR-2 domain–mediated Cdc42 activation for immune regulation.

## Results

### DOCK8 binds specifically to PI(4,5)P2 through the DHR-1 domain

To examine the binding specificity of DOCK8 for phosphoinositides, we used lipid-binding assays in which lysates of cells expressing DOCK8 were incubated with lipid vesicles (1:1 mixture of

phosphatidylethanolamine [PE] and phosphatidylcholine [PC]) containing each phosphoinositide. Lipid vesicles were pelleted, and analyzed by immunoblotting to detect a bound fraction of DOCK8. Specifically, BW5147α⁻β⁻ mouse thymoma cell line that lacks endogenous expression of DOCK8 (Harada et al, 2012) was stably transfected to express HA-tagged DOCK8, and used as an input. As shown in Fig 1A, DOCK8 binds to PI(4,5)P2 prominently, and to much lesser extent to PI(3,4,5)P3, but no such binding was detected with other phosphoinositides.

The DHR-1 domain of the DOCK-A subfamily members, DOCK1 and DOCK2, mediates the interaction with PI(3,4,5)P3 (Côté et al, 2005; Kunisaki et al, 2006). To test whether the DOCK8 DHR-1 domain is responsible for PI(4,5)P2 binding, we obtained BW5147α⁻β⁻ cells expressing a mutant form of DOCK8 that lacks the DHR-1 domain (DOCK8 ΔDHR-1; Fig 1B). Unlike wild-type DOCK8, DOCK8 ΔDHR-1 failed to bind to PI(4,5)P2 (Fig 1C), indicating that the DHR-1 domain is solely responsible for the interaction with PI(4,5)P2. Consistently, the isolated DOCK8 DHR-1 domain expressed as a recombinant GST-fusion protein was sufficient for binding to PI(4,5)P2 (Fig 1D), albeit less prominently than the PLCδ1 PH (pleckstrin homology) domain, a representative of a high affinity PI(4,5)P2–binding domain (Lemmon et al, 1995). These results indicate that DOCK8 binds specifically to PI(4,5)P2 through the DHR-1 domain.

### Crystal structure of the DOCK8 DHR-1 domain

To understand the structural basis for the specific recognition of PI(4,5)P2 by the DOCK8 DHR-1 domain, we performed X-ray crystallographic analysis, and determined its structure at 1.5-Å resolution (Fig 2A and Table 1). The crystal contained one protein molecule per asymmetric unit. The DOCK8 DHR-1 domain consists of a core structure adopting an anti-parallel β-sandwich with a C2 domain fold composed of a pair of four-stranded β-sheets (Fig 2A and C), having a concave face called "β-groove" among C2 domains (Cho & Stahelin, 2006). The core is elaborated with two insertions between β2-β3, and β7-β8, and three loops (L1 through L3) that form a positively charged pocket on the upper surface (Fig 2C and E). The overall structure of the DOCK8 DHR-1 domain resembles the structure of the DOCK1 DHR-1 domain (Fig 2B; PDB ID: 3L4C; Premkumar et al, 2010) with a root mean square deviation of 3.7 Å over 149 Cα atoms, despite low sequence identity (21%). Notable distinction is found in the conformation of L1 to L3 loops and electrostatic surface potential of the upper surface pocket (Fig 2A–F). In the DOCK8 DHR-1 domain, L1 and L2 loops mainly create the upper surface pocket (Fig 2E), whereas all three loops participate in the pocket in the DOCK1 DHR-1 domain (Fig 2F). In addition, the electrostatic surface potential of the upper surface pocket is less prominent in the DOCK8 DHR-1 domain.

To gain further insight into phospholipid binding, we solved the structure of the DOCK8 DHR-1 domain crystalized in the presence of diC8-PI(4,5)P2 (0.84 mM) at 1.43-Å resolution (Fig 2G and Table 1). Although we were unsuccessful in visualizing PI(4,5)P2 moiety, the structure of DOCK8 DHR-1 domain is distinct from its "free" form (Fig 2A), markedly in the upper surface pocket (Figs 2H and S1). Strikingly, the orientation of the side chains of three basic residues (R570 on β1 strand; K576 and R581 in L1 loop) are shifted inward of the pocket as an additional short helix is formed around R581 in the

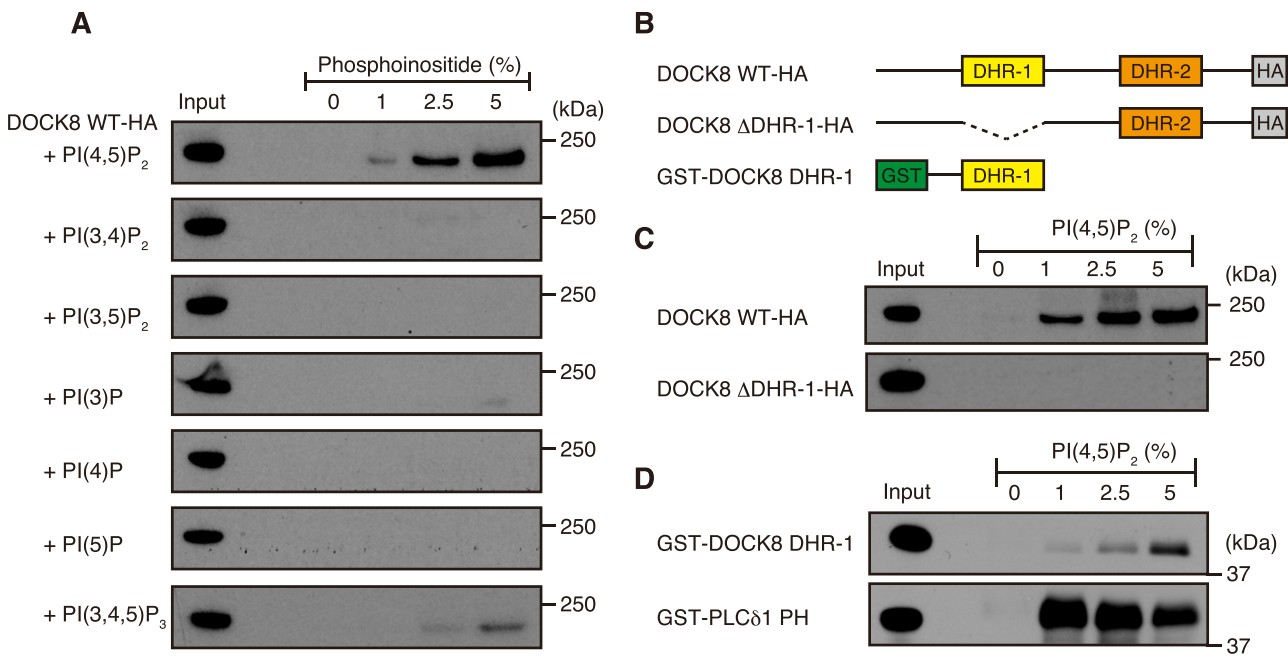

**Figure 1. DOCK8 binds to PI(4,5)P2 specifically through the DHR-1 domain.**
**(A)** Immunoblots showing the phosphoinositide-binding specificity of DOCK8. Lysates of BW5147α⁻β⁻ cells expressing HA-tagged wild-type (WT) DOCK8 were used as an input for incubation with lipid vesicles containing each phosphoinositide at indicated concentrations (%). Lipid-associated fractions were analyzed by immunoblotting with anti-HA antibody. Positions of the size marker were shown on the right. **(B)** Schematic diagram of DOCK8 constructs used in the experiments. **(C)** Immunoblots showing no detectable binding to PI(4,5)P2 of a DOCK8 mutant in which the DHR-1 domain is deleted (ΔDHR-1). Lysates of BW5147α⁻β⁻ cells expressing HA-tagged WT DOCK8 (top) or DOCK8-ΔDHR-1 (bottom) were used as an input for lipid-binding assays. **(D)** Immunoblots showing PI(4,5)P2 binding of recombinant GST-fusion DOCK8 DHR-1 domain (top) and GST-fusion PLCδ1 PH domain (bottom).

presence of diC8-PI(4,5)P2, suggesting that these residues are involved in PI(4,5)P2 binding.

### Residues in the upper surface pocket critical for specific recognition of PI(4,5)P2

Based on the structural information, we chose key candidate residues within the upper surface pocket for specific recognition of PI(4,5)P2, and introduced their point mutations to GST-fusion DOCK8 DHR-1 domain (Fig 3). Specifically, six residues R570 (β1 strand), N572/K576/R581/N582 (L1 loop), and H622 (L2 loop) were substituted with alanine, and examined for their effect on PI(4,5)P2 binding in lipid-binding assays. As shown in Fig 3A, single point mutations at either lysine 576 (K576A) or arginine 581 (R581A) completely abolished PI(4,5)P2 binding as did their doubly mutant (K576A/R581A; designated as "KARA"), excluding any role played by the GST moiety in PI(4,5)P2 binding in the assays. Mutation at arginine 570 (R570A) also led to a significant loss of PI(4,5)P2 binding, whereas the other mutations N572A, N582A, and H622A did not affect the binding significantly.

To confirm the results, we used isothermal titration calorimetry (ITC), and precisely measured the binding affinity of the DOCK8 DHR-1 domain for diC8-PI(4,5)P2 in solution (Fig 3B and C). Data show that diC8-PI(4,5)P2 bound to wild-type DOCK8 DHR-1 domain with 19.5 ± 3.7 µM affinity in approximately a 1:1 stoichiometry. This affinity is 6.5 times lower than the affinity of the DOCK1 DHR-1 for PI(3,4,5)P3 measured by ITC under similar

conditions (Kd = 3.0 ± 0.9 µM; Premkumar et al, 2010). The binding of PI(4,5)P2 to the DOCK8 DHR-1 domain was largely entropy-driven unlike PI(3,4,5)P3 binding to the DOCK1 DHR-1 domain, which is largely enthalpy-driven. Mutations R570A, K576A, and R581A led to two to threefold reduction in the binding affinity, with R581A mutant exhibiting the lowest affinity (64.2 ± 7.7 µM), suggesting that R581 is the most critical for PI(4,5)P2 binding. Consistently, no binding was observed for KARA mutant. ITC experiments also confirmed the weak PI(3,4,5)P3 binding by DOCK8 DHR-1 with 26.9 ± 3.0 µM affinity, which was also canceled by KARA mutation (Fig S2).

To further assess the effect of KARA mutation in the context of full-length DOCK8, we obtained BW5147α⁻β⁻ cells expressing DOCK8 KARA mutant and examined its binding to PI(4,5)P2-containing vesicles (Fig 3D). The result shows that KARA mutation completely abolished the ability of DOCK8 to bind PI(4,5)P2, indicating that K576 and R581 play essential roles in PI(4,5)P2 recognition by DOCK8.

### Model for PI(4,5)P2 binding to the DOCK8 DHR-1 domain

Taking the experimental data into consideration, we computationally calculated a docking model for the DOCK8 DHR-1·diC8-PI(4,5)P2 complex (Fig 4A–D) based on the crystal structure of the DOCK8 DHR-1 domain in the presence of diC8-PI(4,5)P2 (Fig 2G). In the model, side chains of R570, K576, and R581 form multiple hydrogen bonds with the phosphate

**Table 1.  Crystallographic statistics.**

| Data collection | SAD | Native-free form | Native + diC8-PI(4,5)P2 |
|---|---|---|---|
| Beamline | BL26B2, SPring-8 | BL26B2, SPring-8 | BL26B2, SPring-8 |
| Wavelength (Å) | 0.9790 | 1.0 | 1.0 |
| Space group | $P4_12_12$ | $P4_12_12$ | $P4_12_12$ |
| Cell dimensions | | | |
| $a, b, c$ (Å) | 89.4, 89.4, 48.6 | 89.3, 89.3, 49.5 | 89.0, 89.0, 49.4 |
| $\alpha, \beta, \gamma$ (°) | 90, 90, 90 | 90, 90, 90 | 90, 90, 90 |
| Resolution range (Å) | 50–1.55 (1.61–1.55) | 50–1.50 (1.55–1.50) | 50–1.43 (1.48–1.43) |
| Redundancy | 14.5 | 12.9 | 12.9 |
| Unique reflections | 29,327 (2,860) | 32,687 (3,182) | 37,170 (3,588) |
| Completeness (%) | 99.9 (100.0) | 99.9 (99.8) | 99.8 (98.4) |
| $I/\sigma(I)$ | 25.5 (3.4) | 35.8 (3.0) | 37.0 (3.3) |
| $R_{meas}$[a] | 0.086 (0.793) | 0.073 (0.886) | 0.070 (0.963) |
| Phasing | | | |
| No. of Se sites | 2 | | |
| FOM | 0.464 | | |
| Refinement | | | |
| Resolution range (Å) | | 33.15–1.50 (1.54–1.50) | 33.08–1.43 (3.27–1.43) |
| No. of reflections | | 32,634 | 74,567 |
| $R$-factor/free $R$-factor[b] | | 0.1753/0.1986 | 0.1957/0.2187 |
| No. of atoms | | | |
| Protein | | 1,439 | 1,430 |
| Water | | 234 | 197 |
| Average B values (Å$^2$) | | | |
| Protein | | 28.1 | 29.9 |
| Water | | 36.4 | 37.9 |
| Root-mean-square deviations | | | |
| Bond lengths (Å) | | 0.005 | 0.005 |
| Bond angles (°) | | 0.848 | 0.818 |
| Ramachandran plot (%) | | | |
| Favored | | 90.4 | 89.7 |
| Allowed | | 8.3 | 9.0 |
| Generously allowed | | 0.6 | 0.6 |
| Outliers | | 0.6 | 0.6 |

All numbers in parentheses refer to the highest resolution shell statistics.
[a]$R_{meas}$ = $\Sigma|I_i - I_{avg}|/\Sigma I_i$, where $I_i$ and $I_{avg}$ are the observed and average intensities.
[b]Free $R$ factor was calculated for 5% of randomly selected reflections excluded from refinement.

groups at 1, 4, and 5 positions of the inositol ring for stereo-specific recognition of PI(4,5)P2 (Fig 4A and B). PI(4,5)P2 moiety is well fitted in the basic pocket (Fig 4C and D). The model explains the preference of DOCK8 for PI(4,5)P2 over PI(3,4,5)P3 as the residues N572 and F573 in L1 loop will collide with the phosphate group at 3 position of PI(3,4,5)P3. Thus, two basic residues in L1 loop and additional one on β1 strand of DOCK8 DHR-1 participate in PI(4,5)P2 recognition in sharp contrast to the reported model

of DOCK1 DHR-1–binding PI(3,4,5)P3, where three basic resdiues in L1 loop and one in L3 loop are involved (Premkumar et al, 2010).

Among the DOCK-C subfamily, DOCK6 and DOCK7 contain Ser, Arg, and Arg at positions corresponding to DOCK8 R570, K576, and R581, respectively (Fig S3). Our simulation to dock a DOCK8 R570S mutant with diC8-PI(4,5)P2 resulted in formation of a hydrogen bond between S570 and the phosphate group at one position of the inositol

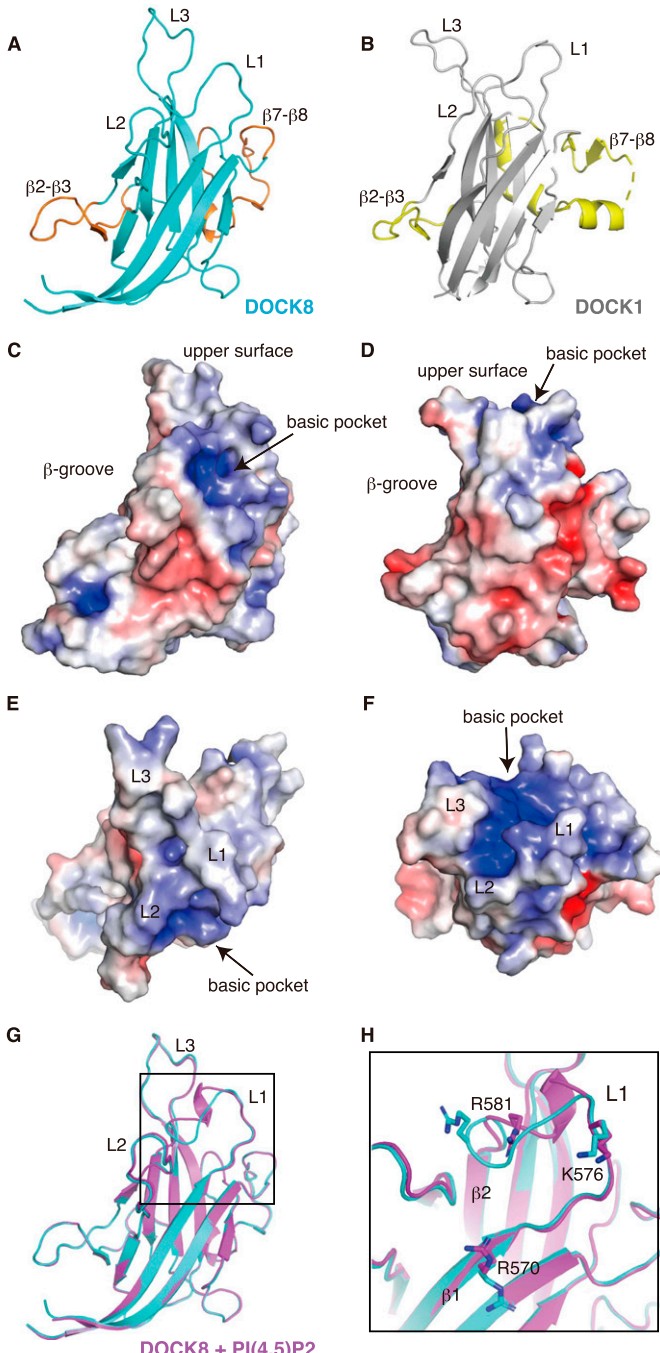

**Figure 2. Crystal structure of the DOCK8 DHR-1 domain.**
**(A)** Ribbon diagram of the DOCK8 DHR-1 domain. The β-sandwich core is colored in cyan; β2-β3 loop and β7-β8 insertion are in orange. Three loops on the upper surface are labeled with numbers (L1 through L3). **(B)** Ribbon diagram of the DOCK1 DHR-1 domain (PDB ID: 3L4C; gray and light-yellow). **(C, D, E, F)** Surface charge representation of the DOCK8 DHR-1 domain (C, E) and the DOCK1 DHR-1 domain (D, F). Blue and red represent positive and negative electrostatic potential, respectively. The views in (C, D) are the same as those in (A, B); their top views are shown in (E, F), respectively. The upper surface region, β-groove, and basic pocket are indicated. **(G)** Ribbon diagram of the DOCK8 DHR-1 domain in the presence of 0.84 mM diC8-PI(4,5)P2 (purple) superposed with the one in the absence of PI(4,5)P2 shown in (A) (cyan). **(H)** Close-up view of the upper surface pocket of the DOCK8 DHR-1 domain. The boxed region in (G) is slightly tilted for the frontal view of the L1 loop region. Residues R570, K576, and R581 are highlighted by a stick model to show the different conformations.

ring (Fig S4). Thus, it is likely that DOCK6 and DOCK7 can bind PI(4,5)P2 in a similar mode to DOCK8.

## The DHR-1 domain is critical for plasma membrane (PM) targeting of DOCK8 and its ability to facilitate 3D cell migration

Having resolved the structural basis for specific recognition of PI(4,5)P2 by the DOCK8 DHR-1 domain, we next examined its role in DOCK8 functions. PI(4,5)P2 is generated by phosphatidylinositol 4-phosphate 5-kinase from PI(4)P, and most enriched in the plasma membrane in mammalian cells (Di Paolo & De Camilli, 2006; Tuosto et al, 2015). Our previous study has shown that DOCK8 accumulates preferentially at the plasma membrane, and controls localized Cdc42 activation at the leading edge of migrating DCs and macrophages (Harada et al, 2012; Shiraishi et al, 2017). To examine the role of the DHR-1 domain in plasma membrane localization of DOCK8, cellular localization of various forms of DOCK8 in BW5147α⁻β⁻ cells were examined by immunofluorescence microscopy. As previously shown, a significant fraction of wild-type DOCK8 localized to the plasma membrane, and colocalized with WGA lectin staining (Fig 5A and B). In contrast, DOCK8 KARA and ΔDHR-1 mutants exhibited primarily diffused distribution in the cytoplasm. Quantification in multiple cells revealed that the level of DOCK8 WT was highest at the plasma membrane, whereas the level of KARA or ΔDHR-1 was higher in the regions distant from the plasma membrane (Fig 5C). Thus, the PM to Cyto ratio within 1 μm of the plasma membrane was significantly higher for DOCK8 WT (1.47 ± 0.75) compared with KARA, or ΔDHR-1 (0.91 ± 0.31, or 0.90 ± 0.27, respectively) (Fig 5D). Consistent results were obtained when the subcellular localization of DOCK8 was analyzed biochemically (Fig S5). These results indicate that the DHR-1 domain is essential for the plasma membrane localization of DOCK8.

Next, we examined whether the DHR-1 domain is required for DOCK8-mediated Cdc42 activation. Live cell imaging with a fluorescence resonance energy transfer (FRET)-based biosensor for active Cdc42, Raichu-Cdc42, was used to monitor cellular Cdc42 activity (Aoki & Matsuda, 2009). Specifically, COS-7 cells were transiently transfected with the Raichu-Cdc42 expression vector, and control or the respective DOCK8 constructs. As shown in Fig 5E, FRET efficiency (relative emission ratio of YFP to CFP) reflecting cellular Cdc42 activity was increased to 1.12 ± 0.13 by expression of DOCK8 WT, whereas no such increase was observed for KARA or ΔDHR-1 mutant (1.00 ± 0.09, or 1.00 ± 0.07, respectively). Thus, the DHR-1 domain is required for DOCK8-mediated Cdc42 activation.

Then, we examined the effect on 3D cell migration. Parental BW5147α⁻β⁻ cells, which lacks endogenous expression of DOCK2 and DOCK8, are immotile in 3D collagen gels, and expression of both DOCK2 and DOCK8 markedly increased their random motility (Harada et al, 2012; Fig 5F). This is because DOCK2-mediated Rac activation confers the driving force for movement, whereas DOCK8-mediated Cdc42 activation is required for coordination of actin protrusions. On the other hand, cells expressing DOCK2 and DOCK8 ΔDHR-1 or KARA mutant failed to migrate efficiently in collagen gels (Fig 5F), suggesting that DOCK8 mutants defective in PI(4,5)P2 binding, despite having intact DHR-2 domain, cannot support localized Cdc42 activation during 3D cell migration. Of note, the cells expressing the DOCK8 mutants exhibited significantly lower motility compared with cells expressing only DOCK2; there could be involved some dominant-negative effect of overexpression of mutant

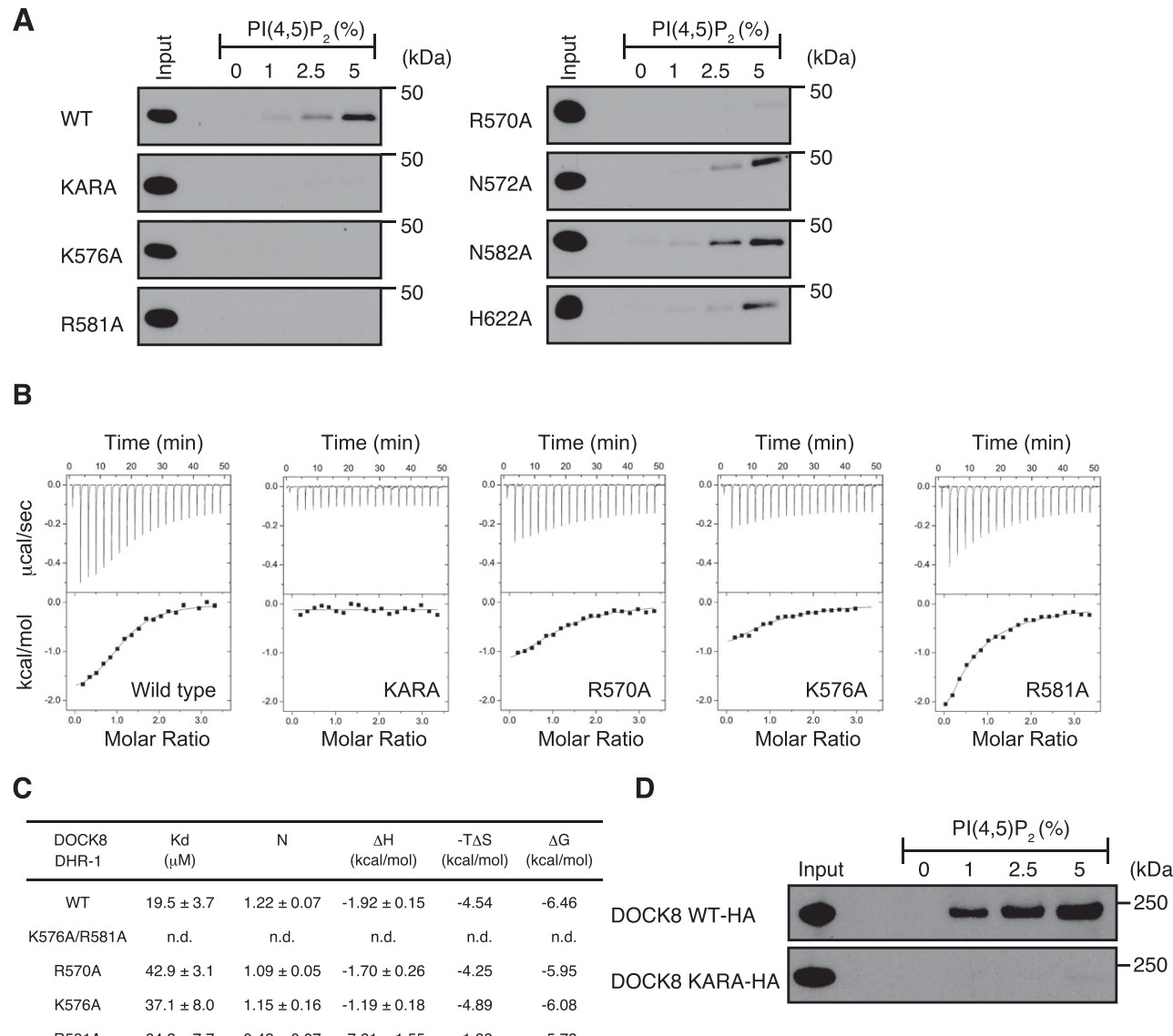

**Figure 3. Identification of critical residues in the DOCK8 DHR-1 domain essential for PI(4,5)P2 binding.**
**(A)** Immunoblots showing the effect of point mutations on PI(4,5)P2 binding of GST-fusion DOCK8 DHR-1 domain. Mutant proteins carry alanine substitution at the indicated residue. A doubly mutant carrying K576A and R581A is designated as "KARA." **(B, C)** Measurement of PI(4,5)P2 binding to DOCK8 DHR-1 by isothermal titration calorimetry. Conditions: 0.062 mM DOCK8 DHR-1 protein titrated with 2 μl aliquotes of 1 mM diC8-PI(4,5)P2 in 20 mM Tris (pH 8.0) and 16 mM NaCl at 25°C. **(B)** Representative titration plots for each DOCK8 DHR-1 (n = 3). Data were best fitted to acquire the stoichiometry and thermodynamic parameters. **(C)** Summary of the experiments. Kd: dissociation constant; ΔH: enthalpy change; TΔS: temperature (K) x entropy change; N: stoichiometry. Data were expressed as means ± SD (n = 3). **(D)** Immunoblots showing no detectable binding of DOCK8 KARA mutant to PI(4,5)P2. Lysates of BW5147α⁻β⁻ cells expressing HA-tagged WT DOCK8 (top) or DOCK8 KARA (bottom) were used as an input for lipid-binding assays.

proteins. Taken together, these results demonstrate that the DHR-1 domain is essential for the plasma membrane localization of DOCK8 and its ability to activate Cdc42 and facilitate 3D cell migration.

### DOCK8 KARA mutation attenuates DC migration in 3D microenvironments

DCs are the most potent antigen-presenting cells that stimulate T cells for antigen stimulated immune responses. DOCK8 deficiency

impairs their interstitial migration in tissues, leading to defective T-cell priming in vivo (Harada et al, 2012). To address the functional significance of PI(4,5)P2 binding by the DOCK8 DHR-1 domain in a physiological context, we generated knock-in mice harboring DOCK8 KARA mutation (K576A/R581A) using the clustered regulatory interspaced short palindromic repeat (CRISPR)/CRISPR–associated 9 (Cas9) system-mediated gene editing. The KARA mutant and wild-type mice were crossed with DOCK8 deficient null (*DOCK8⁻/⁻*) mice to obtain heterozygotes *DOCK8^KARA/⁻* and *DOCK8^WT/⁻* mice. BM DCs were prepared from each strain, and analyzed by immunoblotting

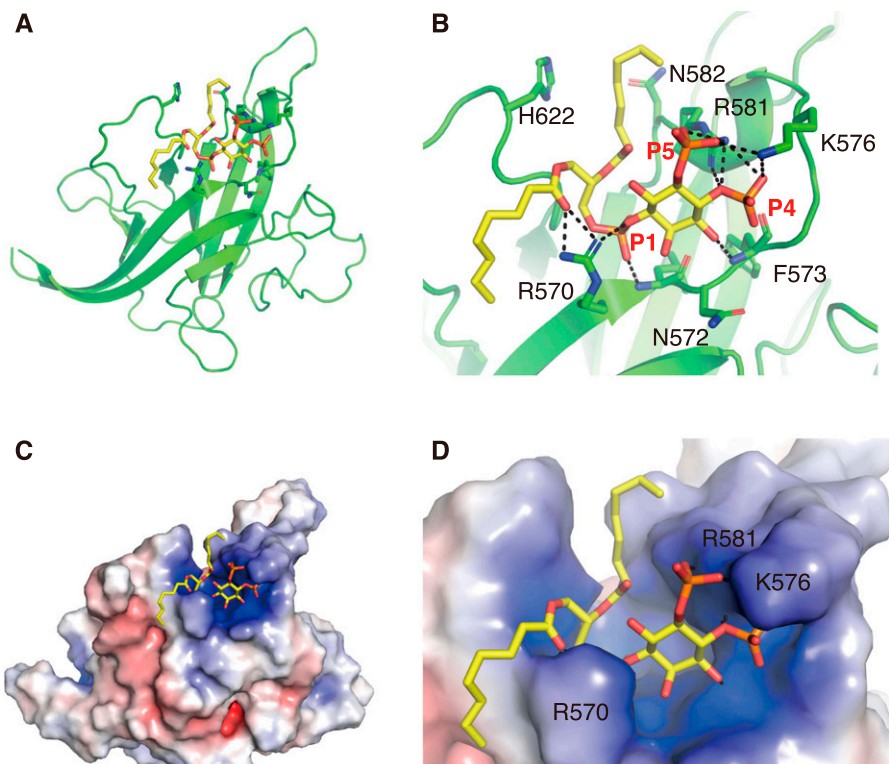

**Figure 4. Model for PI(4,5)P2 binding to the DOCK8 DHR-1 domain.**
**(A)** Images of diC8-PI(4,5)P2 docked into the upper surface pocket of the DOCK8 DHR-1 domain. PI(4,5)P2 are highlighted by a stick model. DOCK8 DHR-1 is shown in ribbon diagram (green). **(B)** Close-up view of the upper surface pocket in (A). Residues predicted to form the pocket and/or bind phospholipid are shown. **(C)** Surface charge representation of DOCK8 DHR-1 in (A) showing the electrostatic surface of the phospholipid-binding pocket. **(D)** Close-up view of the binding pocket in (C).

for the expression level of DOCK8 protein (Fig 6A). KARA mutant protein was expressed at a comparable level to wild DOCK8 protein. When placed under a gradient of chemokine CCL21 in collagen gels, LPS-stimulated, mature WT (*DOCK8^WT/−*) DCs migrated efficiently toward the source of the gradient with an average speed of 1.36 $\mu$m/min (Fig 6B and C). In contrast, *DOCK8^KARA/−* DCs did not migrate as efficiently; *DOCK8^KARA/−* DCs migrated at 0.66 $\mu$m/min, 1/2 speed of WT DCs, yet significantly faster than *DOCK8^−/−* DCs (0.40 $\mu$m/min; *P* = 0.0002). Directionality (straightness of the path) was also impaired in *DOCK8^KARA/−* DCs to the extent similar to *DOCK8^−/−* DCs, and forward migration index (movement toward the chemotactic source) of *DOCK8^KARA/−* DCs was slightly better than *DOCK8^−/−* DCs (Fig 6C). A special type of leukocyte cell death during migration in 3D environments has been recently reported, and termed as cytothripsis (Zhang et al, 2014). However, the extent of cell death under the experimental conditions (1.65 mg/ml collagen) was comparable among the DCs (average 17.4%, 16.7%, and 21.0% apoptotic cells for WT, *DOCK8^KARA/−*, and *DOCK8^−/−* DCs, n = 2). The expression of CCR7 (the receptor for CCL21) was also comparable among the DCs of three genotypes (Fig S6). When WT DCs and *DOCK8^KARA/−* or *DOCK8^−/−* DCs were labeled with different dyes, mixed at 1:1 ratio, and injected into the footpads of C57BL/6 mice, the migration efficiency of *DOCK8^KARA/−* DCs into the popliteal LNs was reduced to 50% of WT DCs, but three times better than that of *DOCK8^−/−* DCs (Fig 6D). Thus, KARA mutation significantly attenuated DC migration in 3D environments. These results demonstrate that PI(4,5)P2 binding

through the DHR-1 domain is critical for immune regulation by DOCK8.

## Discussion

Phosphoinositide binding has an exquisite regulatory role in signaling and functions of proteins at the membrane-cytosol interface (Di Paolo & De Camilli, 2006; Tuosto et al, 2015). Yet, the phosphoinositide binding specificity of the DOCK family of Rac/Cdc42 GEFs is not comprehensively understood. In the present study, we have shown that the DOCK8 DHR-1 domain binds PI(4,5)P2 specifically, and identified key residues R570/K576/R581 through structural, biochemical, and mutational analyses. Our docking model accommodates the residues R570, K576, and R581 of the DOCK8 DHR-1 domain in positions interacting with phosphates at 1, 4, and 5 positions of the inositol ring for stereospecific recognition of PI(4,5)P2. The three basic residues are evolutionarily conserved in DOCK8 (Fig S3). Among the DOCK-C subfamily, DOCK6 and DOCK7 contain Ser, Arg, and Arg at the corresponding positions, which are compatible with the same binding mode, consistently with the previous result showing that DOCK7 as well as DOCK8 preferentially binds to PI(4,5)P2 in pull down assays with PIP beads (Jungmichel et al, 2014).

The DHR-1 domain is required for plasma membrane targeting of DOCK8, and its ability to activate Cdc42 and support leukocyte 3D

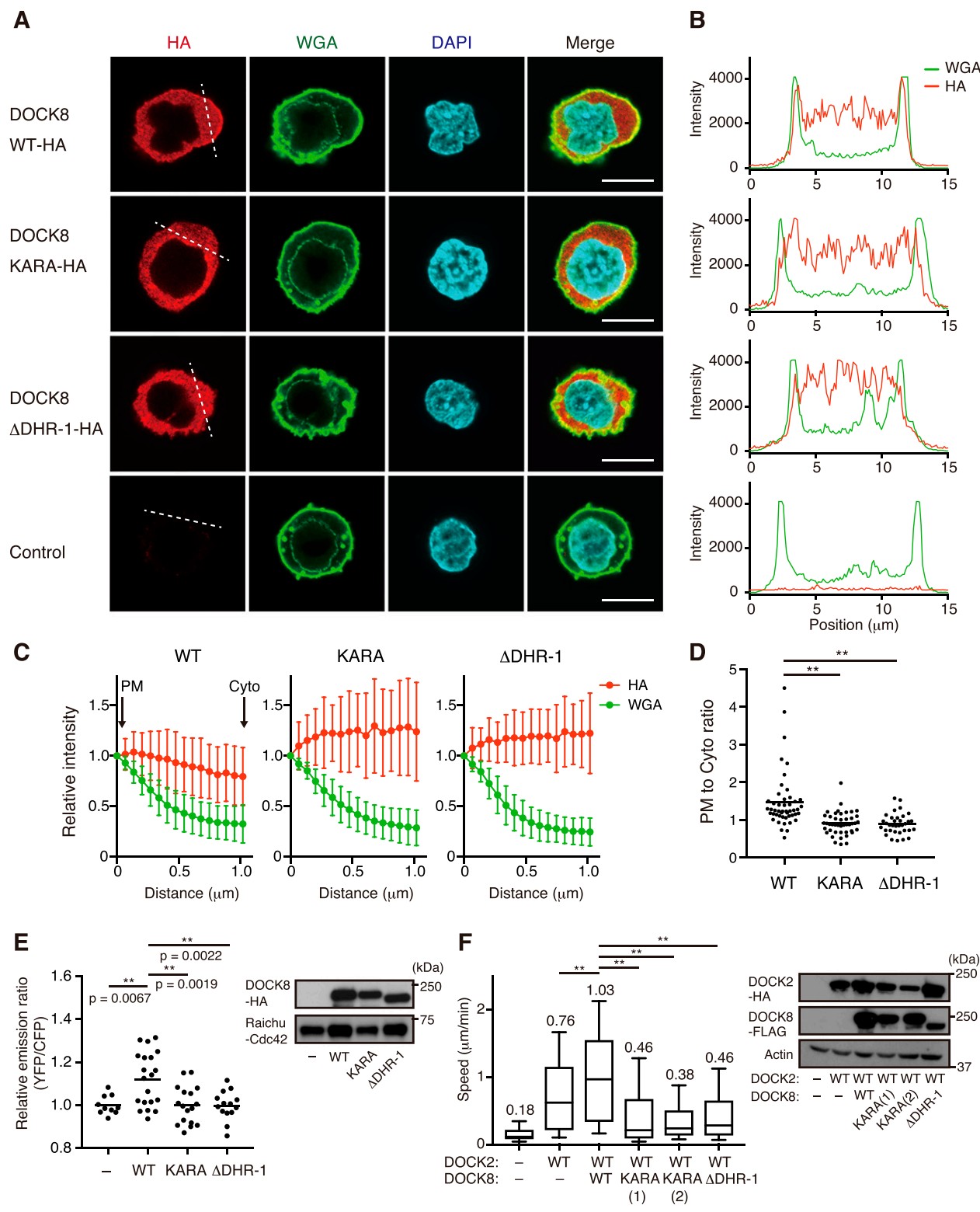

**Figure 5. DHR-1 domain is required for plasma membrane (PM) targeting of DOCK8 and its ability to activate Cdc42 and facilitate 3D cell migration in collagen gels.**
**(A)** Confocal images showing the cellular localization of HA-tagged WT and mutant DOCK8. BW5147$\alpha^-\beta^-$ cells stably expressing indicated proteins were analyzed by immunofluorescence using anti-HA antibody (red). Alexa Fluor 488–conjugated WGA (green) and DAPI (4′,6-diamidino-2-phenylindole, blue) were used to stain the cell surface membrane and nucleus, respectively. Scale bar: 10 $\mu$m. **(B)** Line scanned intensity profiles for HA and WGA fluorescence in the respective cells depicted in (A). Fluorescence intensity of HA and WGA stainings was scanned along the dotted lines in (A). The x-axis indicates arbitrary position on the line. **(C)** Quantification of PM localization of WT and mutant DOCK8. The intensity profiles for HA (red) and WGA (green) fluorescence from mupltiple cells were averaged (n = 48/24, 44/22, and 34/18

migration. By generating novel knock-in mice in which two key residues for PI(4,5)P2 binding are mutated (K576A/R581A), we show that the DHR-1 domain is critical for immune regulatory functions of DOCK8. This is the first report of such revealing the functional significance of a DHR-1 domain in a physiological context. Our data agree with the loss of function phenotypes of some DOCK8 deficient patients whose *DOCK8* gene contains intact, apparently functional DHR-2 GEF domain, still missing a large portion of the N-terminal region including the entire DHR-1 domain (Engelhardt et al, 2009).

Our results uncover a critical role of DOCK8 in coupling PI(4,5)P2 signaling with Cdc42 activation through its DHR-1 and DHR-2 domains, respectively. Leukocyte migration in 3D environments depends on DOCK8-mediated localized Cdc42 activation for leading edge coordination (Lämmermann et al, 2009; Harada et al, 2012). The fact that DOCK8 DHR-1 mutants devoid of PI(4,5)P2 binding cannot support efficient 3D migration suggests a strong link between PI(4,5)P2 signal and localized activation of Cdc42. Wiskott-Aldrich syndrome protein (WASP) is a Cdc42 effector that stimulates Arp2/3 complex–mediated actin polymerization, and promotes formation of filopodial protrusions (Takenawa & Suetsugu, 2007). Like DOCK8, WASP is critical for DC migration and T-cell priming in vivo (Snapper et al, 2005; Bouma et al, 2007), and its deficiency leads to immunodeficiency in humans (Ochs & Thrasher, 2006). Remarkably, PI(4,5)P2 and GTP-loaded Cdc42 are the two coincident signals that activate WASP synergistically by liberating its closed, inactive conformation for an active conformation (Prehoda et al, 2000; Takenawa & Suetsugu, 2007), the significance of which was first hinted by the work on actin assembly in Xenopus egg extracts (Ma et al, 1998). Thus, a Cdc42 GEF (DOCK8) and its effector (WASP) functionally converge through PI(4,5)P2 (Fig 7). Analogously, the DOCK-A subfamily members bind PI(3,4,5)P3, and activate Rac, and in turn, PI(3,4,5)P3 and GTP-Rac activate the WASP-family verprolin-homologous protein (WAVE) family proteins through destabilizing its inhibitory protein complex (Takenawa & Suetsugu, 2007). Activated WAVEs stimulate Arp2/3 complex and promote lamellipodial protrusions to drive cell motility (Oikawa et al, 2004). Not appreciated by far, the intimately-coupled relationships between the DOCK family and WASP/WAVE family proteins may imply a biological significance of their intricate, concerted actions in signal integration.

A high affinity for PI(3,4,5)P3 of the DOCK-A subfamily (Kd ~3 $\mu$M) would be necessary for its recruitment to a membrane microdomain containing PI(3,4,5)P3, which is transiently generated by active PI 3-kinase at the cell leading edge. On the other hand, much lower affinity of the DOCK8 DHR-1 domain for PI(4,5)P2 (Kd ~19 $\mu$M)

may well be suited for sampling a membrane domain, acting as a coincidence detector of PI(4,5)P2 and GDP-Cdc42. Whereas the low affinity could be sufficient for targeting DOCK8 to the plasma membrane, where PI(4,5)P2 is present at estimated 10–24 $\mu$M (Lemmon et al, 1995; McLaughlin et al, 2002), a 10–20 times higher levels than PI(3,4,5)P3 (Tuosto et al, 2015), it is possible that protein dimerization (Yang et al, 2009; Terasawa et al, 2012) and interaction with other binding partners increase the avidity in the cells. For example, DOCK8 directly interacts with LRAP35a, an adaptor protein that binds to a Cdc42 effector myotonic dystrophy kinase-related Cdc42-binding kinase (MRCK) (Shiraishi et al, 2017). This interaction is essential for linking DOCK8-mediated Cdc42 activation to actomyosin dynamics during leukocyte migration. Also, DOCK8 associates with MST1 (Yamamura et al, 2017), a Ste20-like serine/threonine kinase that is critical for lymphocyte trafficking and interstitial motility (Katagiri et al, 2009). In budding yeast, Ste20p is a Cdc42p effector that functions as a part of the cell polarity determinant machinery (Hall, 1998; Etienne-Manneville, 2004; Wedlich-Soldner & Li, 2008). There are also reports on the association of DOCK8 with WASP in a macromolecular complex (Ham et al, 2013; Janssen et al, 2016). Thus, DOCK8 does not only act as an upstream regulator of Cdc42 but also interacts with PI(4,5)P2 and a number of signaling proteins involved in the polarization and actin reorganization. In this regard, it is noteworthy that the budding yeast scaffold protein Bem1p brings together Cdc24p (a Cdc42 GEF), Cdc42p, and downstream effectors of Cdc42p to form a positive feedback loop that stabilizes the Cdc42 GEF at sites of polarization (Butty et al, 2002) for avidity-driven polarity establishment (Meca et al, 2019). Similar feedback mechanism may operate for DOCK8. In summary, we have shown here that DOCK8 links PI(4,5)P2 signaling to Cdc42 activation in immune cell regulations. Understanding the molecular network and signal integration mechanism of DOCK8 should provide a mechanistic insight into the spatiotemporal control of Cdc42 activity during interstitial leukocyte migration.

# Materials and Methods

### Cell culture and transfections

BW5147$\alpha^-\beta^-$ mouse thymoma cell line (White et al, 1989), and its stable transfectants were maintained in RPMI 1640 medium (FUJIFILM Wako Pure Chemical Corporation) supplemented with 10% FBS, 100 U/ml penicillin, and 100 $\mu$g/ml streptomycin (all from Life Technologies).

---

regions/cells for DOCK8 WT, KARA, and ΔDHR-1, respectively). Data are means ± SD. Positions of the peak intensity of WGA fluorescence were defined as the PM, and the fluorescence intensity at PM was set as 1.0 for normalization of HA and WGA fluorescence in individual cells. Distance from PM was plotted on x-axis. **(D)** Plasma membrane to cytoplasmic ratio of HA fluorescence for WT and mutant DOCK8 protein. The ratio of HA fluorescence intensity at the PM and cytoplasm (Cyto) in (C) was plotted for individual cells. **P < 0.0001 by a two-tailed unpaired Mann-Whitney test. **(E)** Fluorescence resonance energy transfer (FRET)–based measurement of Cdc42 activity in living cells. COS-7 cells were co-transfected with a FRET-based biosensor (Raichu-Cdc42), and the pBJ-neo or the respective DOCK8 constructs. FRET imaging was performed during 26–32 h after transfection. Relative emission ratio (YFP/CFP) of the whole cell area was calculated (n = 10, 20, 17, and 14 cells from three independent experiments for control, WT, KARA, and ΔDHR-1, respectively). *P*-values by a two-tailed unpaired *t* test. Right panel: Immunoblots showing the expression of transfected DOCK8 and Raichu-Cdc42 (probed with anti-HA and anti-GFP antibodies, respectively). **(F)** 3D migration in collagen gels of BW5147$\alpha^-\beta^-$ cells expressing HA-tagged DOCK2 and FLAG-tagged WT or mutant DOCK8 (n = 232–286 cells per group from three independent experiments). Two independent clones were analyzed for KARA mutation. Each box plot indicates the median (the line in the middle), 25th and 75th percentiles (box ends), and 10th and 90th percentiles (whiskers). The number on each column indicates the average speed in $\mu$m/min. **P < 0.0001 by a two-tailed unpaired Mann–Whitney test. Right panel: immunoblots showing the expression level of DOCK2 and DOCK8 in the cells. Actin blot is shown as a loading control; the positions for the size markers on the right.

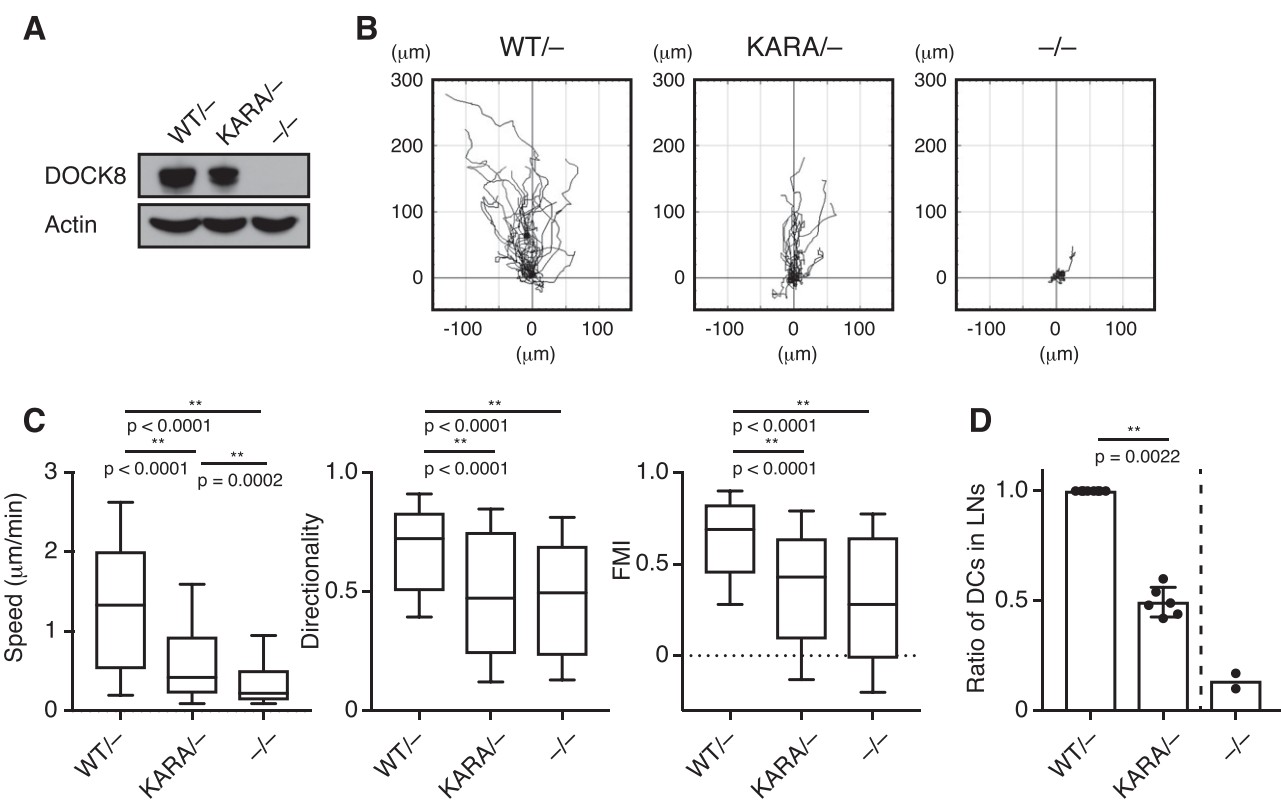

**Figure 6. KARA mutation in DOCK8 significantly attenuates 3D migration of mDCs.**
**(A)** Immunoblot analysis of the expression level of DOCK8 protein in DCs derived from *DOCK8*[WT/−], *DOCK8*[KARA/−], and *DOCK8*[−/−] mice. Actin blot is shown as a loading control. The positions for the size markers were indicated on the right. **(B)** Impaired migration of *DOCK8*[KARA/−] and *DOCK8*[−/−] mature DCs (mDCs) in 3D collagen gels. Migration of LPS-stimulated *DOCK8*[WT/−], *DOCK8*[KARA/−], and *DOCK8*[−/−] mDCs toward CCL21 source was recorded for 120 min by time-lapse video microscopy. Representative tracks of individual mDCs. **(C)** The migration speed, directionality, and forward migration index were compared among *DOCK8*[WT/−], *DOCK8*[KARA/−], and *DOCK8*[−/−] mDCs (n = 132 (109), 127 (75), and 100 (41), respectively, from three independent experiments). For directionality and forward migration index, the cells that had migrated at 0.3 $\mu$m/min or faster were analyzed (cell numbers in the parentheses). Each box plot indicates the median (the line in the middle), 25th and 75th percentiles (box ends), and 10th and 90th percentiles (whiskers). *P*-values by a two-tailed unpaired Mann–Whitney test. **(D)** In vivo migration efficiency of *DOCK8*[WT/−], *DOCK8*[KARA/−], and *DOCK8*[−/−] mDCs. DCs in a pair were mixed at 1:1 ratio, injected into footpads of C57BL/6 mice and recovered from the popliteal LNs after 48 h. Data are means ± SD for six pairs of *DOCK8*[WT/−] and *DOCK8*[KARA/−] DCs with data for two pairs of *DOCK8*[WT/−] and *DOCK8*[−/−] DCs. *P*-value by a two-tailed unpaired Mann–Whitney test.

BW5147α⁻β⁻ cells expressing HA-tagged DOCK8, DOCK8 ΔDHR-1 (residues 561–738 deleted), or KARA mutant were generated by electroporation of the respective pBJ-neo–based construct, and clonal selection with G418 at 2 mg/ml. BW5147α⁻β⁻ cells expressing HA-tagged DOCK2 and FLAG-tagged DOCK8, DOCK8 ΔDHR-1, or KARA mutant were generated by electroporation of the respective pSI-hygro-based construct into BW5147α⁻β⁻ cells expressing HA-tagged DOCK2 (Harada et al, 2012), and clonal selection with hygromycin at 1 mg/ml.

COS-7 cells were maintained in DMEM (FUJIFILM Wako Pure Chemical Corporation) supplemented with 10% FBS, 100 U/ml penicillin, and 100 $\mu$g/ml streptomycin.

### Protein expression and purification

The genes encoding the mouse DOCK8 DHR-1 domain (residues 558–740) and human PLCδ1 PH domain (residues 1–170) were cloned in pGEX6P-1 vector (GE Healthcare), and expressed as a GST-fusion protein in *Escherichia coli* ArcticExpress DE3 strain (Agilent Technologies) by induction with 0.5 mM IPTG at 16°C overnight. Protein was purified by affinity chromatography on Glutathione Sepharose 4B beads (GE

Healthcare). PCR-directed mutagenesis was used to introduce point mutations in the DOCK8 DHR-1 construct. For ITC experiments, the N-terminal GST-tag was removed by digestion with PreScission protease.

For structural analysis, the gene encoding the DOCK8 DHR-1 domain (residues 556–740) was cloned into the pCR2.1 vector (Invitrogen) with an N-terminal His-tag and a tobacco etch virus protease cleavage site. Selenomethionine (SeMet)-labeled and non-labeled proteins were synthesized by the large-scale dialysis mode of the *E. coli* cell-free reaction (Terada et al, 2014; Katsura et al, 2017). Each protein was purified by His-tag affinity chromatography. After digestion with tobacco etch virus protease, the protein was further purified by size-exclusion chromatography on a HiLoad 16/60 Superdex 75 pg column (GE Healthcare Life Sciences).

### Lipid-binding assays

Lipid vesicles were prepared as follows. Briefly, an equal weight mixture of L-α-phosphatidylcholine (PC;840051P; Avanti) and 3-sn-phosphatidylethanolamine (PE; #P4264; Sigma-Aldrich) supplemented with specified concentrations (w/w) of each diC8-phosphoinositide

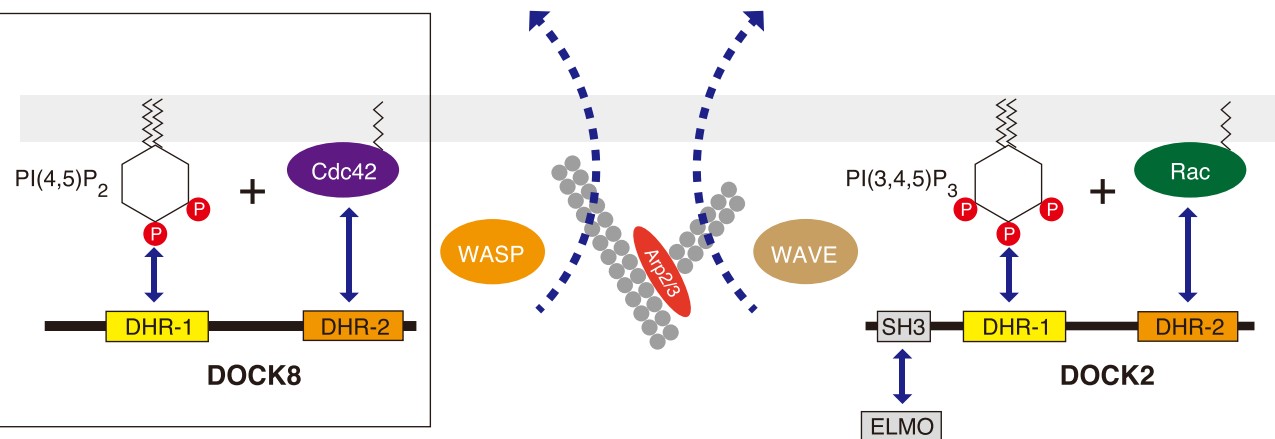

**Figure 7.   Schematic illustrating critical roles of the DOCK family proteins in linking phosphoinositide signaling to specific Rho-Wiskott-Aldrich syndrome protein family pairs.**
Through the DHR-1 domain, DOCK8 is localized to a PI(4,5)P2–enriched compartment of the plasma membrane, where, upon encounter with GDP-Cdc42, the DHR-2 domain catalyzes the nucleotide exchange reaction of Cdc42 (the current work marked in a box). GTP-loaded Cdc42 and PI(4,5)P2 serve as the coincidence detection signals for activation of Wiskott-Aldrich syndrome protein, which stimulates Arp2/3 complex–mediated actin polymerization for reorganization of the actin cytoskeleton. In a similar, but distinct way, the DOCK-A subfamily member (e.g., DOCK2), which makes a signaling complex with engulfment and cell motility (ELMO) protein through the SH3 domain (Hanawa-Suetsugu et al, 2012; Chang et al, 2020), is recruited to the leading edge of migrating cells through the PI(3,4,5)P3–binding DHR-1 domain in response to chemoattractant signals. The DHR-2 domain of the DOCK-A subfamily activates Rac, which in turn acts in synergy with PI(3,4,5)P3 to activate WAVE complex. See text for details.

(Avanti 840150P, 840151P, 840152P, 840153P, 840154P, 840155P, and 840156P) dissolved in chloroform (0.8 mg total lipid in 80–100 $\mu$l) were dried with the Speed Vac concentrator (Savant Instruments). Lipids were resuspended in TBS (20 mM Tris–HCl, 0.15 M NaCl, pH 7.5), and incubated in a water bath at 37°C for 30 min. The solution was mixed vigorously by vortex for 10 min to obtain lipid vesicles. Before use, lipid vesicles were precipitated at 20,000$g$ for 5 min, washed twice with TBS, and resuspended with 200 $\mu$l of binding buffer (20 mM Tris–HCl, 0.15 M NaCl, 1 mM EDTA, pH 7.5). Lysates of BW5147$\alpha^-\beta^-$ cells were prepared by resuspending cells in binding buffer supplemented with 0.2% NP-40 and 1 mM PMSF at 2.0 × 10$^6$ cells/300 $\mu$l, and going through freeze-thawing cycles using liquid nitrogen and water bath at 37°C. Lysates were clarified by centri-fugation at 20,000$g$ for 5 min at 4°C. 25 $\mu$l of clarified lysates was mixed with 100 $\mu$l of lipid vesicles (0.4 mg total lipid) in 1 ml of binding buffer supplemented with 1 mM PMSF, and incubated for 90 min at 4°C on a rotating wheel. Lipid vesicles were pelleted by centrifugation at 20,000$g$ at 4°C, and washed twice with ice-cold binding buffer supplemented with 0.004% NP-40 and 1 mM PMSF. The pellets were mixed with an equal volume of 2× sample buffer (125 mM Tris–HCl, 4% SDS, 20% glycerol, 0.01% bromophenol blue, and 5% mercaptoethanol, pH 6.8), and boiled for 5 min. The fraction associated with lipid vesicles was separated by SDS–PAGE and immunoblotted using anti-HA antibody to detect DOCK8 protein. For the analyses of the isolated DHR-1 domain proteins, lipid vesicles were resuspended with binding buffer supplemented with 0.1% fatty acid free BSA (#A8806; Sigma-Aldrich), and 1 $\mu$g of each GST-fusion protein was used with binding buffer supplemented with 0.006% NP-40. When comparing with the PLC$\delta$1-PH domain, PE was replaced with L-$\alpha$-phosphatidylinositol (PI; #P0639; Sigma-Aldrich) as previously reported (Oikawa et al, 2004).

## Crystallization and structure analysis

The SeMet-labeled DOCK8 DHR-1 crystals were grown at 20°C using the hanging-drop vapor diffusion method by mixing the protein solution (12 mg/ml in 20 mM Tris–HCl at pH 8.0, 150 mM NaCl, and 2 mM DTT) with an equal volume of reservoir solution (22% PEG3350 and 0.2 M potassium chloride). The single-wavelength anomalous dispersion (SAD) data were collected at 100 K with a wavelength of 0.9790 Å at BL26B2, SPring-8. Similarly, high-resolution native datasets were collected at a wavelength of 1.0 Å with non-labeled DOCK8 DHR-1 crystals grown with and without 0.84 mM diC8-PI(4,5)P2 (P-4508; Echelon). The crystallization conditions for the non-labeled DOCK8 DHR-1 crystals were 20% PEG3350 and 0.2 M di-sodium hydrogen phosphate in the absence of diC8-PI(4,5)P2, and 20% PEG3350 and 0.2 M sodium sulfate decahydrate in the presence of diC8-PI(4,5)P2. The diffraction data were processed with the HKL2000 program (Otwinowski & Minor, 1997). The structure of DOCK8 DHR-1 was solved by SAD phasing with SHELX (Sheldrick, 2010) and AutoSol (Terwilliger et al, 2009) programs, and the native datasets diffracting to higher resolution were used to refine the model. The model was corrected iteratively using the program Coot (Emsley & Cowtan, 2004) and refined using PHENIX (Adams et al, 2010). The quality of the model was inspected by PROCHECK (Laskowski et al, 1993). Molecular graphics were generated using PyMOL (Delano, 2002).

## Modeling

The AutoDock Vina program (Trott & Olson, 2010) was used to perform molecular docking. The structure of DOCK8 DHR-1 crys-talized with 0.84 mM diC8-PI(4,5)P2 was used to dock diC8-PI(4,5)P2.

Binding poses were searched within a grid box enclosing two critical PI(4,5)P2–binding residues, K576 and R581, in DOCK8. The docked structure was further minimized using the Amber simulation programs (Case et al, 2014). The structure of the DOCK8 DHR-1 R570S mutant was created using Coot, docked with diC8-PI(4,5)P2, and minimized similarly to wild-type DOCK8 DHR-1.

## ITC

ITC measurements were carried out on a MicroCal Auto-iTC200 (Malvern Instruments) at 25°C. Each titration was performed with 19 injections of 2 $\mu$l aliquots of 1 mM diC8-PI(4,5)P2 or diC8-PI(3,4,5)P3 (both from Echelon Biosciences) at 150 s intervals into a reaction cell containing 62 $\mu$M DOCK8 DHR-1 or its mutants. The buffer consisted of 20 mM Tris–HCl (pH 8.0) and 16 mM NaCl. Data were analyzed using the Origin 7 software (OriginLab Corporation).

## Immunofluorescence microscopy

Cells were seeded on 35-mm poly-L-lysine-coated glass-bottom dishes (D11131H; Matsunami Glass), and cultured overnight. Then, cells were washed once with PBS, and fixed with 4% paraformaldehyde/PBS. Fixed cells were washed three times with PBS, and incubated with Alexa Fluor 488–conjugated WGA (final 3–10 $\mu$g/ml, W11261; Invitrogen) for 10 min at RT. Cells were permeablilized for 5 min with 0.1% Triton-X-100/PBS, and blocked with 1% BSA/PBS for 1 h at RT. Cells were incubated with anti-HA antibody (1:5,000, clone 3F10; Roche) for 1 h at RT, and with Alexa Fluor 546–conjugated goat anti-rat IgG antibody (1:2,000, A11081; Invitrogen) for 1 h at RT. Cells were then stained with DAPI (1:5,000; WAKO), and mounted using the fluorescent mounting medium (DAKO). Images were taken with a confocal laser scanning microscope FV3000 (Olympus Corporation) using a UPlanSApo 60×/1.35 na oil immersion objective lenz (Olympus Corporation). For quantification of colocalization, images were analyzed by the line profile function of the built-in cellSens application. For each cell, a well-defined area of cytoplasm to plasma membrane was identified, and intensity profile along a single traverse line was obtained. Points with peak fluorescence intensity of WGA staining on both ends of the cell periphery were defined as the PM, and set as zero distance, and intensity profile of HA-staining over 1.1 $\mu$m from PM were analyzed.

## Subcellular fractionation

BW5147$\alpha^-\beta^-$ cells (1 × 10$^7$ cells) were resuspended in 700 $\mu$l of ice-cold hypotonic buffer (42 mM KCl, 10 mM Hepes [pH 7.4], 5 mM MgCl$_2$) supplemented with 1× complete protease inhibitor cocktail (Roche), and incubated on ice for 15 min. Cells were transferred to a 1 ml syringe, and sheared by passing through a 27 gauge neddle five times. The lysates were centrifuged at 200$g$ for 10 min to remove nuclei and cell debris. Then, the lysates were centrifuged at 13,000$g$ for 60 min at 4°C, and the supernatant was collected as the "cytoplasmic fraction." The pellet was resuspended in lysis buffer (20 mM Tris–HCl [pH 7.5], 150 mM NaCl, 1% NP-40) supplemented with 1× complete protease inhibitor cocktail, and detergent-extracted by vortexing, and incubating on a rotating wheel for 45 min at 4°C.

Then, the sample was centrifuged at 13,000$g$ for 60 min at 4°C, and the supernatant was collected as the "membrane fraction."

## FRET imaging

Live cell imaging with a FRET-based biosensor Raichu-Cdc42 was used to measure Cdc42 activity in living cells (Aoki & Matsuda, 2009). COS-7 cells were seeded in 60 mm dishes, and transfected with 0.5 $\mu$g of the Raichu-Cdc42 expression plasmid (1054X) and 4.5 $\mu$g of pBJ-neo or the respective DOCK8 construct using Lipofectamine 2000 transfection reagent (Invitrogen). 16 h after transfection, the cells were reseeded onto fibronectin (F1141; Sigma-Aldrich)-coated 35-mm glass-bottom dishes (D11130H; Matsunami Glass). The culture medium was replaced with fresh phenol red-free RPMI 1640 medium (Gibco) supplemented with 20 mM Hepes (pH 7.3) before microscopic analysis. FRET imaging was performed on the Olympus IX-81 fluorescence microscope equipped with an ORCA-Flash 4.0 digital CMOS camera (Hamamatsu) and a UPlanSApo 60×/1.35 na oil immersion objective lenz (Olympus Corporation) in a dark room warmed at 30°C. The culture dishes were placed on a heated stage set at 37°C, and kept for 2 min before imaging for 5 min with 30 s intervals with excitation using an LED illumination precisExcite fluorescence excitation system (CoolLED), excitation filter 440AF21 (Omega), and a 455DRLP dichroic mirror (Omega), and emission filter 480AF30 (Omega) with exposure time 250 ms for CFP (excitation at 480 nm), and emission filter 535AF26 (Omega) with exposure time 500 ms for YFP (reporting FRET; excitation at 555 nm). Using the MetaMorph software, background was subtracted from each stack of the raw CFP and YFP images. The average intensity of CFP and YFP in the whole area of transfected cells was measured, and the values of YFP to CFP emission ratio over 2–5 min were averaged, and normalized to the values of control cells (transfected with pBJ-neo) in each experiment.

## Immunoblotting analysis

Immunoblotting was performed with following antibodies: rat monoclonal antibody for HA (1:2,000 dilution, 3F10; Roche), anti-GST antibody (1:500, 013-21851; Wako), rabbit anti-GFP antibody (1:1,000, A11122; Invitrogen), mouse anti-Cdc42 antibody (1:1,000, 05-542; Millipore), rabbit anti-LAT antibody (1:1,000 06-807; Millipore), HRP-conjugated rabbit anti-FLAG (DDDDK-tag) antibody (1:2,000, PM020-7; MBL), custom-made rabbit anti-DOCK8 antibody (1:1,000; Harada et al, 2012), goat anti-actin (1:1,000, sc-1616; Santa Cruz), and corresponding species-specific HRP-conjugated anti-IgG antibodies (1:2,000; all from Santa Cruz). Blots were developed on Super RX X-ray films (Fujifilm) with the ECL or ECL Prime western blotting detection reagents (GE Healthcare).

## Mice

All mice were on genetic background of C57BL/6J strain (CLEA Japan). DOCK8 null mice were described previously (Harada et al, 2012). Mice harboring K576A/R581A mutation in *DOCK8* were generated by CRISPR/Cas9–mediated genome editing as described below. Mice were maintained under specific pathogen-free conditions in the animal facility of Kyushu University. The

protocol of animal experiments was approved by the committee of Ethics on Animal Experiments, Kyushu University.

## CRISPR/Cas9–mediated genome editing in mouse pronuclear zygote

A custom-designed DOCK8 locus-specific crRNA (5′-GCUGGGG-GUAUACGUACAGAGUUUUAGAGCUAUGCU-3′; Target guide RNA sequence on the DOCK8 exon 15 underlined; Integrated DNA Technologies) was made into duplex with the generic tracrRNA (#1072532; Integrated DNA Technologies) in Nuclease-Free Duplex Buffer (30 mM Hepes, pH 7.5, 100 mM potassium acetate). 60 pmol of the duplex guide RNA was incubated with 6 $\mu$g of Cas9-3NLS protein (Integrated DNA Technologies) in 1× Opti-MEM (Thermo Fisher Scientific) for 10 min at RT to form a Cas9-guide RNA (RNP) complex, then followed by addition of 180 pmole of single-stranded donor oligo (5′-CTTACCGGCATAGCATTGCTGGGGTCTTCTCCGCACATAAACTGAATCTTTA TTGTGATGTTCGCGGCAGATGCTAGCGCGCTAGCGAAGTTCAGTCGCTGGGGG TAGACATACAGAAGGTTTCTGGAAAAGAGAGAGACCAATGATCAACAGATGATG TCTGGAGTTCACTTCGGCTTT-3′; mutations corresponding to K576A/R581A, double-underlined; silent mutations blocking Cas9 re-cutting, underlined) as a template for homology-directed DNA repair. Mouse pronuclear-stage embryos were obtained by the standard in vitro fertilization method from C57BL/6J donors (CLEA Japan). At 7 h from insemination, the Cas9 RNP complex and donor oligo were transferred to intact pronuclear-stage embryos using a NEPA 21 super electroporator and 5 mm-gap platinum metal electrode (NEPA Gene). Set parameters were as follows. Poring pulse: voltage 225 V, pulse length 2 ms, pulse interval 50 ms, number of pulses +4, and decay rate 10%. Transfer pulse: voltage 20 V, pulse length 50 ms, pulse interval 50 ms, number of pulses ± 5, and decay rate 40%. The embryos were cultured overnight, and two-cell stage embryos were selected and transferred to ICR host female mice (Charles River Laboratories). To identify correctly targeted founder mice, the DOCK8 locus spanning exon 15 was amplified by PCR and their sequence was verified. The founder mice were crossed with C57BL/6 males to obtain progenies with transmitted locus.

## Preparation of DCs

To generate BM-derived DCs, BM cells were cultured for 7 d with granulocyte-macrophage colony-stimulating factor (GM-CSF; 10 ng/ml; PeproTech), and purified with anti-CD11c microbeads (130-108-338; Miltenyi Biotec). Purified DCs were treated with LPS (200 ng/ml; Sigma-Aldrich) for 24 h for maturation before assays.

## 3D-cell migration assays

For DCs, 3D collagen gel chemotaxis assays were performed with $\mu$-Slide Chemotaxis3D (ibidi; Harada et al, 2012). Cells were mixed with 1.65 mg/ml of collagen type I (354236; Corning) in phenol red-free RPMI 1640 (11835-030; Gibco) supplemented with 4% FBS, and cast in the central chamber. After polymerization of the lattice, the media with or without mouse CCL21 (10 $\mu$g/ml, 457-6C-025; R&D Systems) were added to the wells on either side of the chamber to form a gradient between them. Images were taken in a heated chamber every 2 min using an IX-81 inverted microscope (Olympus

Corporation) equipped with a cooled CCD camera (CoolSNAP HQ; Roper Scientific), an IX2- ZDC laser-based autofocusing system (Olympus Corporation), and an MD-XY60100T-Meta automatically programmable XY stage (Sigma KOKI). Images were imported as stacks to ImageJ Version 1.410 software and analyzed with the manual tracking and the chemotaxis and migration tools to measure the migration speed, directionality, and forward migration index. Migration of BW5147$\alpha^-\beta^-$ cells was analyzed by placing collagen-cell mixture on 35-mm glass-bottomed microwell dishes (P35G-0-10-C; MatTek), followed by incubation in phenol red-free RPMI supplemented with 10% FBS. Images were taken, and velocities were calculated with the manual tracking feature in MetaMorph software.

## Reverse transcription-qPCR

Total RNA was isolated from LPS-stimulated DCs using ISOGEN reagent (Nippon Gene), and treated with RNase-free DNase I (Invitrogen). RNA samples were reverse transcribed with oligo(dT)-20 primers (TOYOBO) and Superscript III reverse transcriptase (Invitrogen). Real-time quantitative PCR was performed on a CFX Connect real-time system (Bio-Rad) using the SYBR Green PCR Master Mix (Applied Biosystems) with primers: Ccr7-forward: CCCAGAGCACC-ATGGACCC, and Ccr7-reverse: CTCGTACAGGGTGTAGTCCACC; and Gapdh forward: GGAGAAACCTGCCAAGTATGATG, and Gapdh reverse: AAGA-GTGGGAGTTGCTGTTGAAG. Expression of *Ccr7* was normalized to the expression of *Gapdh* for each sample.

## Flow cytometry

LPS-stimulated DCs were preincubated with rat anti-CD16/32 (Fc$\gamma$III/II receptor; 1:1,000, 2.4G2; TONBO Biosciences) for 10 min on ice to block Fc receptors, and then stained with PE-conjugated rat anti-mouse CCR7 (1:10, 4B12; BioLegend) or PE-conjugated isotype-matched control antibody (rat IgG2a, $\kappa$; 1:10, RTK2758; BioLegend) for 30 min at 37°C. Flow cytometric analyses were performed on a FACS Calibur (BD Biosciences).

## Cell viability in collagen gels

LPS-stimulated DCs were mixed with collagen type I (0, 1.65, and 3.3 mg/ml) in phenol red-free RPMI 1640 supplemented with 4% FBS, seeded in 24-well plates, and incubated for 30 min at 37°C to solidify the gel. Then, cells were overlaid with the media containing CCL21 at 10 $\mu$g/ml to let undergo random migration for 60 min at 37°C. Then, cells were recovered from the gels by incubating with 0.1% collagenase type I. Cells were stained with propidium iodide (PI) and FITC-conjugated Annexin V, and analyzed by flow cytometry for early apoptotic (PI$^-$, Annexin V$^+$) and late apoptotic cells (PI$^+$, Annexin V$^+$).

## In vivo DC migration assays

LPS-stimulated *DOCK8*$^{WT/-}$ DCs and *DOCK8*$^{KARA/-}$ or *DOCK8*$^{-/-}$ DCs were labeled with PKH26 and PKH67 fluorescent dyes (Sigma-Aldrich), respectively, resuspended in 5 × 10$^5$ cells in 20 $\mu$l, and mixed at 1:1. Total one million cells in 40 $\mu$l were injected

subcutaneously into the hind footpads of C57BL/6 mice. After 48 h, cells were recovered from the popliteal LNs, and analyzed by flow cytometry on a FACSVerse (BD Biosciences).

## Statistical analyses

Data were examined for normal distribution by a Kolmogorov-Smirnov test using the built-in function of GraphPad Prism 7 software. Parametric data were analyzed by a two-tailed unpaired $t$ test, and nonparametric data were analyzed by a two-tailed unpaired Mann–Whitney test. $P < 0.05$ was considered statistically significant. $P < 0.01$ was indicated by **.

# Data Availability

The X-ray crystal structures reported here have been deposited in the Protein Data Bank with accession codes 7CLX and 7CLY.

# Supplementary Information

# Acknowledgements

We thank Y Tanaka of Fukuoka Dental College, and F Sanematsu of Shokei University for their expert advice and guidance on confocal imaging, and A Inayoshi, A Aosaka, N Kanematsu, and M Tanaka of Kyushu University for their technical assistance. We also thank C Mishima-Tsumagari, K Katsura, and T Nishizaki of RIKEN for their technical assistance in protein preparation, the beamline staff at SPring-8 for assistance with data collection, and K Tsuda of RIKEN for his advice on molecular docking. The synchrotron radiation experiments at SPring-8 were partially performed with the approval of the Japan Synchrotron Radiation Research Institute (JASRI) (Proposal No. 2012A1211). We would like to thank A Shimada of Kyushu University for critical reading of the manuscript. This research was supported by Leading Advanced Projects for Medical Innovation (LEAP, JP19gm0010001), Advanced Research and Development Programs for Medical Innovation (AMED-CREST and JP20gm1310005), and Practical Resarch Project for Allergic Diseases and Immunology (JP20ek0410064) from Japan Agency for Medical Research and Development (AMED) (to Y Fukui); and Grants-in-Aid for Scientific Research from Ministry of Education, Culture, Sports, Science and Technology of Japan (to Y Fukui, M Kukimoto-Niino, and T Uruno).

## Author Contributions

T Sakura: investigation and writing—original draft, review, and editing.

M Kukimoto-Niino: formal analysis, funding acquisition, investigation, and writing—original draft, review, and editing.

K Kunimura: investigation.

N Yamane: investigation.

D Sakata: investigation.

R Aihara: investigation.

T Yasuda: methodology.

S Yokoyama: investigation.

M Shirouzu: investigation.

Y Fukui: conceptualization, funding acquisition, and supervision.

T Uruno: formal analysis, supervision, funding acquisition, validation, investigation, project administration, and writing—original draft, review, and editing.

## Conflict of Interest Statement

The authors declare that they have no conflict of interest.

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
