## [Reviewer comments · Life Science Alliance]

Life Science Alliance

A conserved PI(4,5)P2-binding domain is critical for immune regulatory function of DOCK8

Tetsuya Sakurai, Mutsuko Kukimoto-Niino, Kazufumi Kunimura, Nana Yamane, Daiji Sakata, Ryosuke Aihara, Tomoharu Yasuda, Shigeyuki Yokoyama, Mikako Shirouzu, Yoshinori Fukui, and Takehito Uruno

DOI: <https://doi.org/10.26508/lsa.202000873>

Corresponding author(s): Takehito Uruno, Medical Institute of Bioregulation, Kyushu University

Review Timeline:

Submission Date:	2020-08-12
Editorial Decision:	2020-09-08
Revision Received:	2020-12-06
Editorial Decision:	2021-01-19
Revision Received:	2021-01-20
Accepted:	2021-01-20

Scientific Editor: Shachi Bhatt

Transaction Report:

September 8, 2020

Re: Life Science Alliance manuscript #LSA-2020-00873-T

Dr. Takehito Uruno
Medical Institute of Bioregulation, Kyushu University
Division of Immunogenetics, Department of Immunobiology and Neuroscience
3-1-1 Maidashi, Higashi-ku
Fukuoka 812-8582
Japan

Dear Dr. Uruno,

Thank you for submitting your manuscript entitled "An evolutionarily conserved PI(4,5)P₂-binding domain is critical for immune regulatory function of DOCK8" to Life Science Alliance (LSA). The manuscript has been reviewed by the editors and outside referees (reviewer comments below). As you will see, the reviewers were quite enthusiastic about the study and its potential impact, but have raised some concerns that should be addressed prior to further consideration of the manuscript at LSA. Therefore, although we are unable to publish the current version of the manuscript, we would kindly encourage you to submit a revised version that addresses all of the referees' concerns.

We would be happy to discuss the individual revision points further with you should this be helpful. While you are revising your manuscript, please also attend to the below editorial points to help expedite the publication of your manuscript. Please direct any editorial questions to the journal office. The typical timeframe for revisions is three months. Please note that papers are generally considered through only one revision cycle, so strong support from the referees on the revised version is needed for acceptance. When submitting the revision, please include a letter addressing the reviewers' comments point by point.

We hope that the comments below will prove constructive as your work progresses. Thank you for considering LSA as an appropriate venue for your research, we look forward to receiving your revised manuscript.

Sincerely,

Shachi Bhatt, Ph.D.
Executive Editor
Life Science Alliance

B. MANUSCRIPT ORGANIZATION AND FORMATTING:

Reviewer #1 (Comments to the Authors (Required)):

Through structural and biochemical analysis, Sakurai et al. demonstrate the molecular mechanism of DOCK8's (DHR)-1 domain binding to PI(4,5) P2 instead of PI(3,4,5) P3. The authors also show that three basic residues are critical for recognition of PI(4,5) P2 by structural analysis. And substitution of these residues in cell lines or mice ablates the ability of DOCK8 to support cell migration. Overall, this is an elegant study to further clarify the detailed molecular mechanism for the specificity of Dock8 binding to PI(4,5)P2. Although previous work showed that the DHR1 domain of DOCK8 specifically binds to (PI(4,5)P2) and that Dock8 regulates interstitial migration of DCs this work adds new molecular insight into how this is accomplished. Some of the conclusions were not addressed clearly based on the data presented and there are a few experimental questions that need to be addressed as outlined below.

Major comments

1. The paper focused on how Dock8 DHR-1 domain binds to PI(4,5) P2, and to a lesser extent to PI(3,4,5) P3 by structural and biochemical analysis. Previous papers showed that Dock8 activates Cdc42, although weak activity was also reported for Rac (Ham et al., 2013). Does the Dock8 DHR-1 domain binding to PI(4,5) P2 contribute to Cdc42 activation whereas Dock8 DHR-1 domain binding to PI(3,4,5) P3 contributes to Rac activation?
2. Dock6 and 7, which belong to same DOCK-A subfamily with Dock8, activate both CDC42 and Rac. Jugmichel et al in 2014 also showed that Dock7 can bind to PI(4,5) P2, and to a lesser extent to PI(3,4,5) P3. The authors partially showed that Dock6 and 7 still can bind to PI(4,5) P2 although they use different amino acids. What about the binding to PI(3,4,5) P3?
3. In figure 4, the authors used a computational model to show PI(4,5) P2 binding to Dock8 DHR-1 domain and explained the collision (residues N572 and F573) for PI(3,4,5) P3 binding. Direct Data about these would strengthen the manuscript, although not essential. And are the three amino acids, R570, K576 and R581, also critical for PI(3,4,5) P3 binding?
4. In figure 5, did the BW5147 mouse thymoma cell line require a signal to locate Dock8 to the cell membrane?
5. In figure 6, Dock8^{-/-} DCs have little absolute displacement although they have speed, which is different from Dock8 KARA^{-/-} DCs. An explanation for the difference should be given. Can Dock8 KARA can bind to PI(3,4,5) P3 and contribute to the migration difference between Dock8 KARA^{-/-} DCs and Dock8^{-/-} DCs?
6. To further confirm the migration defect, DC migration of DOCK8KARA^{-/-} mice should be analyzed in vivo.

Minor comments

1. In figure 1C, the binding of Dock8 WT-HA with PI(4,5) P2 are not concentration dependent, which is different from figure 1A or 1D. Is there explanation for the difference?

Reviewer #2 (Comments to the Authors (Required)):

This manuscript characterizes the structural binding specificity of the PI(4,5)P2 to DOCK8 and its functional significance. Using crystallographic analysis and computational modelling the authors identify critical residues of DOCK8 DHR-1 domain that are responsible for the specific binding of phosphoinositide PI(4,5)P2. They show that this interaction is key for the recruitment of DOCK8 to the plasma membrane which is critical for efficient cell migration in 3D environments.

Overall the manuscript succeeds in describing the specific interaction between the two molecules and the cellular consequences. The biochemical experiments seem sound, they are well described and performed and are supported in the crystallographic and computational analysis. However, the functional, cell biological experiments are less clear and somewhat preliminary and would need further development.

Although the localization of the different HA constructs seems clear, the immunofluorescence images are of insufficient quality. The variable intensities between images (for example DAPI channel) could suggest that staining protocols require some optimization. Also, a cell type with more cytoplasmic content would improve the visualization of the HA-tagged protein localization.

The use of Concanavalin A as membrane stain requires a step of incubation on ice prior to use, which could disrupt cell cortex and cytoskeleton and affect cell viability. I would recommend to use one of the many commercially available membrane stains that can be directly added to the medium.

Figure 5C shows the different speeds in BW5147 cells expressing DOCK2 and different DOCK8 constructs. Cell expressing DOCK8 mutants that are unable bind the PI(4,5)P2 are significantly slower than DOCK8 KO cells. This is an intriguing result, that should be at least included as part of the discussion, especially as the opposite effect is shown for dendritic cells in Figure 6C, where DOCK8 KARA expressing cells migrate better than DOCK8 KO.

Also, there is no mention about what kind of migration impairment DOCK8 KARA expressing cells display. It is known that DOCK8 KO cells suffer cytothripsis when migrating in 3D environments. It would be important to carefully describe the migration defect in DOCK8 KARA expressing cells and if these cells also suffer cell rupture.

Figure 7

Figure 7 not only summarizes the data from the current manuscript but also includes other information, which would fit more in a review rather than in this manuscript. Especially as the cell type used for these experiments does not show clear lamellipodia or filopodia and DOCK2 is not investigated.

Reviewer #3 (Comments to the Authors (Required)):

The DOCK family of guanine exchange factors (GEFs) differentially promotes the activity of Rho GTPases, Rac and Cdc42. DOCK2 and DOCK8 are critical regulators of immune cell migration, development and differentiation, and whose deficiencies result in distinct human primary immunodeficiency diseases. Considerable biochemical and functional analyses have highlighted important interactions between DOCK8 proteins and various signaling proteins. This group in particular previously determined the structure of DOCK8 bound to Cdc42 and reported its importance in dendritic cell and macrophage migration. In this manuscript, they examine the upstream regulation of DOCK8 activity, specifically how DOCK8 localizes to the membrane during leukocyte migration and activation, which currently remains unclear.

While the field attributes DOCK family membrane localization to the DOCK homology region-1 (DHR-1) domain binding to PI(3,4,5)P3, there has only been one structural study conducted on the association of PIP3 with the DHR-1 of DOCK1, which acts as a GEF for Rac. Given the distinct function and localization of Rac and Cdc42 as well as the non-redundant requirement for DOCK8 in proper immune cell function, the current study identifies important new molecular mechanisms of DOCK8 activity and provides insight into how DOCK8-deficiency results in immune dysfunction. The authors provide convincing biochemical, biophysical and structural data demonstrating specific interactions between DOCK8 and PI(4,5)P2 (19.5 μ M affinity), a phosphoinositide found constitutively in low abundance at the plasma membrane but far more abundant relative to PI3K-inducible PIP3. They identify critical basic residues within the DHR-1 domain that shift orientation

upon PIP2 binding and whose mutations disrupt PIP2 binding. Additionally, reconstitution of a DOCK8-deficient lymphocyte cell line and DOCK8-deficient mice results in defective membrane localization and 3D migration of dendritic cells in response to the chemokine CCL21. I have only minor/moderate concerns that need addressing for an otherwise interesting and important study:

1. Moderate: The effect of DHR-1 deletion or mutation on cellular levels of Cdc42-GTP need to be assessed in order to conclude that PIP2-regulation of DOCK8 is important for regulating Cdc42 activity.
2. Moderate: The authors need to determine whether CCR7 expression levels and localization in migrating dendritic cells are similar between WT, DOCK8-deficient and KARA-reconstituted cells. This is particularly important given prior studies indicating an effect of Dock8 deficiency on transcription of important cell surface receptors.
3. Minor: Change green or aqua color in Fig 2G and H to allow better visualization of the conformational differences between PIP2-bound and -unbound DOCK8.
4. Minor: in Fig 3B, some y-axes are missing negative signs
5. Minor: HA and ConA immunofluorescent signal co-localization in Fig 5A and B should be quantified in multiple cells in a non-biased manner.

Responses to the reviewers:

We thank all the Reviewers for their careful reading of the manuscript. We have now revised the manuscript according to their comments and advice. Our responses to the comments are as follows:

Reviewer #1 (Comments to the Authors (Required)):

Major comments

1. The paper focused on how Dock8 DHR-1 domain binds to PI(4,5) P2, and to a lesser extent to PI(3,4,5) P3 by structural and biochemical analysis. Previous papers showed that Dock8 activates Cdc42, although weak activity was also reported for Rac (Ham et al., 2013). Does the Dock8 DHR-1 domain binding to PI(4,5) P2 contribute to Cdc42 activation whereas Dock8 DHR-1 domain binding to PI(3,4,5) P3 contributes to Rac activation?

We performed ITC experiments for PI(3,4,5)P3 binding to DOCK8 DHR-1, and determined its Kd to be 27 μ M (new Fig EV2). This affinity, lower than that of DOCK1-DHR-1 by nearly one-order, may preclude the possibility of DOCK8 recruited to a PI(3,4,5)P3-containing membrane by virtue of its own DHR-1 domain. While we appreciate the possibility of Rac activation by DOCK8 reported in some literature, the intrinsic Rac GEF activity of the DOCK8 DHR-2 domain is essentially negligible compared to its Cdc42 GEF activity (Harada et al, *Blood* 2012). Taken together, we think that it is worth considering, but maybe unlikely that DOCK8 contributes to Rac activation downstream PI(3,4,5)P3 signaling on its own, and that the primary role of DOCK8 is to link PI(4,5)P2 signaling to Cdc42 activation.

2. Dock6 and 7, which belong to same DOCK-A subfamily with Dock8, activate both CDC42 and Rac. Jugmichel et al in 2014 also showed that Dock7 can bind to PI(4,5) P2, and to a lesser extent to PI(3,4,5) P3. The authors partially showed that Dock6 and 7 still can bind to PI(4,5) P2 although they use different amino acids. What about the binding to PI(3,4,5) P3?

The docking model for PI(4,5)P2 binding to DOCK8 DHR-1 domain (Fig 4) can accommodate PI(3,4,5)P3 by slightly changing the conformation of the inositol ring, as shown below.

However, even in this case, the main chain of F573 will still collide with the phosphate at 3 position without further rearrangement of the DHR-1, consistently with the lower affinity. The situation is essentially the same for DOCK8 DHR-1 R570S (Fig EV3), and thereby predicts that DOCK6/7 DHR-1 may also bind PI(3,4,5)P3 albeit with low affinity. However, to make such conclusion, further experimental validations should be needed.

Model for PI(3,4,5)P3 binding to DOCK8 DHR-1

3. In figure 4, the authors used a computational model to show PI(4,5) P2 binding to Dock8 DHR-1 domain and explained the collision (residues N572 and F573) for PI(3,4,5) P3 binding. Direct Data about these would strengthen the manuscript, although not essential. And are the three amino acids, R570, K576 and R581, also critical for PI(3,4,5) P3 binding?

As discussed above, the model for PI(4,5)P2 binding to DOCK8 DHR-1 domain can accommodate PI(3,4,5)P3 in a different conformation. However, even in this case, without further rearrangement, the main chain of F573 will still collide with the phosphate at 3 position. On the other hand, we performed additional ITC experiments, and confirmed that KARA mutation canceled PI(3,4,5)P3 binding as in the case for PI(4,5)P2, indicating that the same residues, K576 and R581, are critical for PI(3,4,5)P3 binding (new Fig EV2).

4. In figure 5, did the BW5147 mouse thymoma cell line require a signal to locate Dock8 to the cell membrane?

In Figure 5F, cells were placed in collagen gels immersed in DMEM-10%FBS, and let undergo random motility. We think that components in FBS and the steady state level of PI(4,5)P2 at the plasma membrane are likely the source of signals locating a portion of DOCK8 and Cdc42 to the cell membrane.

5. In figure 6, Dock8^{-/-} DCs have little absolute displacement although they have speed, which is different from Dock8 KARA^{-/-} DCs. An explanation for the difference should be given. Can Dock8 KARA can bind to PI(3,4,5) P3 and contribute to the migration difference between Dock8 KARA^{-/-} DCs and Dock8^{-/-} DCs?

We have reanalyzed the data for speed, directionality and FMI (new Fig 6C). While all the speed, directionality, and FMI were substantially impaired in DOCK8^{KARA^{-/-}} and DOCK8^{-/-} DCs

compared to DOCK8^{WT/-} DCs, DOCK8^{KARA/-} DCs exhibited significantly higher motility speed, and slightly better FMI compared to DOCK8^{-/-} DCs (page 10). As discussed, our data indicate that KARA mutant is incapable of binding to PI(3,4,5)P3.

6. To further confirm the migration defect, DC migration of DOCK8KARA/- mice should be analyzed in vivo.

Thank you for your suggestion. We have performed in vivo DC migration assays, and the results indicate that the in vivo migration efficiency of DOCK8^{KARA/-} DCs was reduced to 50% of DOCK8^{WT/-} DCs, but three times better than that of DOCK8^{-/-} DCs (new Fig 6D).

Minor comments

1. In figure 1C, the binding of Dock8 WT-HA with PI(4,5) P2 are not concentration dependent, which is different from figure 1A or 1D. Is there explanation for the difference?

The original blots were replaced with new blots from another experiment. We observed occasionally semi-dose dependent binding probably due to variation in lipid vesicle preparations.

Reviewer #2 (Comments to the Authors (Required)):

Figure 5 Although the localization of the different HA constructs seems clear, the immunofluorescence images are of insufficient quality. The variable intensities between images (for example DAPI channel) could suggest that staining protocols require some optimization. Also, a cell type with more cytoplasmic content would improve the visualization of the HA-tagged protein localization.

According to your advice, we have optimized the staining protocol using WGA-lectin, and improved substantially the quality of immunofluorescence images (new Fig 5A). We have now performed quantification of localization of DOCK8 protein in multiple cells (new Fig 5B-D). In addition, instead of analyzing another cell type, we performed biochemical fractionation experiments, the results of which independently confirmed the membrane localization of WT DOCK8, but not KARA or Δ DHR-1 mutant.

The use of Concavalin A as membrane stain requires a step of incubation on ice prior to use, which could disrupt cell cortex and cytoskeleton and affect cell viability. I would recommend to

use one of the many commercially available membrane stains that can be directly added to the medium.

Following your advice, we have changed the protocol, and replaced ConA with WGA, which allowed us to avoid incubation on ice and fix intact cells directly, which improved substantially the image quality.

Figure 5C shows the different speeds in BW5147 cells expressing DOCK2 and different DOCK8 constructs. Cell expressing DOCK8 mutants that are unable bind the PI(4,5)P2 are significantly slower than DOCK8 KO cells. This is an intriguing result, that should be at least included as part of the discussion, especially as the opposite effect is shown for dendritic cells in Figure 6C, where DOCK8 KARA expressing cells migrate better than DOCK8 KO.

Thank you for your comment. As you pointed out, there might be some ‘dominant-negative’ effect from overexpressing mutant proteins in the experiments. Taking your suggestion, we inserted text “Of note, the cells expressing DOCK8 mutants exhibited significantly lower motility compared with cells expressing only DOCK2; there could be involved some dominant-negative effect of overexpression of mutant proteins.” (page-10). However, in general, we think that the defects in DOCK8^{KARA/-} cells are less severe compared to DOCK8^{-/-} cells under physiological conditions. In addition to the DC phenotypes described here, for example, we have recently reported that DOCK8 controls survival of group 3 innate lymphoid cells (ILC3s) in the gut through Cdc42 activation (Aihara *et al*, 2020, *International Immunology* doi: 10.1093/intimm/dxaa066). Consistently with the findings in DCs, the number of ILC3s in the gut (as defined by lineage negative, c-Kit⁺, Sca-1⁻ cell population) was reduced from 48.1% in DOCK8^{WT/-} mice to 10.5%, and 6.2% in DOCK8^{KARA/-} and DOCK8^{-/-} mice, respectively (data not included).

Also, there is no mention about what kind of migration impairment DOCK8 KARA expressing cells display. It is known that DOCK8 KO cells suffer cytothripsis when migrating in 3D enviroments. It would be important to carefully describe the migration defect in DOCK8 KARA expressing cells and if these cells also suffer cell rupture.

Thank you for raising a possibility of ‘cytothripsis’ playing a role in the migration defects. We examined the viability of DCs under conditions with different collagen concentrations (0, 1.65, 3.3 mg/ml) following the experiments in Zhang *et al*, 2014, *J Exp Med*. As shown below, the percentage of early and late apoptotic cells were comparable among the DCs of three genotypes

under all conditions. There may be a tendency for DOCK8^{-/-} DCs to have more late apoptotic cell populations at all collagen concentrations. However, with collagen gels at 1.65 mg/ml, the conditions for the 3D chemotactic assays, there was no obvious difference in cell viability among the DCs.

Besides, we have reanalyzed the 3D chemotaxis data for speed, directionality and FMI (new Fig 6C). While all the speed, directionality, and FMI were substantially impaired in DOCK8^{KARA/+} and DOCK8^{-/-} DCs compared to DOCK8^{WT/+} DCs, DOCK8^{KARA/+} DCs exhibited significantly higher motility speed, and slightly better FMI compared to DOCK8^{-/-} DCs (page 10). Moreover, following Reviewer 3's suggestion, we examined the mRNA and cell surface presentation levels of CCR7, and found no difference among the DCs. Thus, the chemotactic defects of DOCK8^{KARA/+} DCs in 3D environments are likely due to defects in events related to Cdc42-mediated cytoskeletal reorganization.

Figure 7

Figure 7 not only summarizes the data from the current manuscript but also includes other information, which would fit more in a review rather than in this manuscript. Especially as the cell type used for these experiments does not show clear lamellipodia or filopodia and DOCK2 is not investigated.

Following your advice, we removed the cellular contexts and information (lamellipodia or filopodia, etc.), and marked data from the current manuscript in a box (new Fig 7). The authors still wish to keep the scheme relative to DOCK2 to help readers appreciate the implication of our findings.

Reviewer #3 (Comments to the Authors (Required)):

1. Moderate: The effect of DHR-1 deletion or mutation on cellular levels of Cdc42-GTP need to be assessed in order to conclude that PIP2-regulation of DOCK8 is important for regulating Cdc42 activity.

Our initial attempts to assess the cellular level of Cdc42-GTP in BW5147 $\alpha\beta^-$ cells using pull-down assays with PAK-CRIB beads were unsuccessful likely due to transient nature of active Cdc42 in the cells. Instead, we have performed live cell imaging using a FRET-probe for active Cdc42 (Raichu-Cdc42) in COS-7 cells. The results indicate that FRET efficiency (reporting the cellular level of Cdc42 activity) increases with the expression of WT DOCK8, whereas no such increase was observed for KARA or Δ DHR-1 mutant (new Fig 5E). Based on the results, we now conclude that PIP2-regulation of DOCK8 is important for regulating Cdc42 activity.

2. Moderate: The authors need to determine whether CCR7 expression levels and localization in migrating dendritic cells are similar between WT, DOCK8-deficient and KARA-reconstituted cells. This is particularly important given prior studies indicating an effect of Dock8 deficiency on transcription of important cell surface receptors.

We have examined the mRNA expression and cell surface presentation levels of CCR7 in LPS-stimulated DCs (new Fig EV6). The results indicate that mRNA expression and cell surface presentation of CCR7 were comparable among the DCs of three genotypes.

3. Minor: Change green or aqua color in Fig 2G and H to allow better visualization of the conformational differences between PIP2-bound and -unbound DOCK8.

According to your advice, the color for PIP2-bound DOCK8 is changed to purple in new Fig 2G and H.

4. Minor: in Fig 3B, some y-axes are missing negative signs

The errors have been fixed in new Fig 3B.

5. Minor: HA and ConA immunofluorescent signal co-localization in Fig 5A and B should be quantified in multiple cells in a non-biased manner.

We have optimized and improved the staining protocol for colocalization study using WGA (new Fig 5A), and performed quantification in multiple cells (new Fig 5B-D). The results were further supported by biochemical cell fractionation analyses (Fig EV5).

January 19, 2021

RE: Life Science Alliance Manuscript #LSA-2020-00873-TR

Dr. Takehito Uruno
Medical Institute of Bioregulation, Kyushu University
Division of Immunogenetics, Department of Immunobiology and Neuroscience
3-1-1 Maidashi, Higashi-ku
Fukuoka 812-8582
Japan

Dear Dr. Uruno,

Thank you for submitting your revised manuscript entitled "A conserved PI(4,5)P2-binding domain is critical for immune regulatory function of DOCK8". We would be happy to publish your paper in Life Science Alliance pending final revisions necessary to meet our formatting guidelines.

Along with the points listed below, please also attend to the following:

- please make sure that the author order in the manuscript and our system match
- please add ORCID ID for secondary corresponding author-they should have received instructions on how to do so
- please add your supplementary figure legends to the main manuscript text, directly under the main figure legends
- please rename your EV figures as supplementary figures

A. FINAL FILES:

B. MANUSCRIPT ORGANIZATION AND FORMATTING:

Sincerely,

Shachi Bhatt, Ph.D.
Executive Editor
Life Science Alliance
<https://www.lsjournal.org/>
Tweet @SciBhatt @LSAJournal

Reviewer #1 (Comments to the Authors (Required)):

The authors have done impressive work to address the reviewers questions and the revised manuscript is significantly strengthened. The one concern raised by multiple reviewers that was insufficiently addressed was a more molecular discussion of the differences between KARA^{-/-} DCs and Dock8^{-/-} DCs in migration displacement and speed.

Reviewer #3 (Comments to the Authors (Required)):

The authors provide convincing biochemical, biophysical and structural data demonstrating specific interactions between DOCK8 and PI(4,5)P₂, a phosphoinositide found constitutively in low abundance at the plasma membrane. They identify critical basic residues within the DHR-1 domain that shift orientation upon PIP₂ binding and whose mutations disrupt PIP₂ binding. Additionally, reconstitution of a DOCK8-deficient lymphocyte cell line and DOCK8-deficient mice result in defective membrane localization and 3D migration of dendritic cells in response to the chemokine CCL21.

The authors have gone above and beyond my expectations and have thoroughly addressed the minor and moderate concerns I raised. In this revised manuscript, they also demonstrate the importance of DOCK8's PIP₂-binding basic residues in Cdc42 activation.

January 20, 2021

RE: Life Science Alliance Manuscript #LSA-2020-00873-TRR

Dr. Takehito Uruno
Medical Institute of Bioregulation, Kyushu University
Division of Immunogenetics, Department of Immunobiology and Neuroscience
3-1-1 Maidashi, Higashi-ku
Fukuoka 812-8582
Japan

Dear Dr. Uruno,

Thank you for submitting your Research Article entitled "A conserved PI(4,5)P2-binding domain is critical for immune regulatory function of DOCK8". It is a pleasure to let you know that your manuscript is now accepted for publication in Life Science Alliance. Congratulations on this interesting work.

DISTRIBUTION OF MATERIALS:

Again, congratulations on a very nice paper. I hope you found the review process to be constructive and are pleased with how the manuscript was handled editorially. We look forward to future exciting submissions from your lab.

Sincerely,

Shachi Bhatt, Ph.D.

Executive Editor

Life Science Alliance

<https://www.lsjournal.org/>
